# Bootstrap Your Uncertainty: Adaptive Robust Classification Driven by Optimal-Transport

Jiawei Huang[1,2]    Minming Li[2]    Hu Ding[1*]

[1]School of Computer Science and Technology, University of Science and Technology of China
[2]Department of Computer Science, City University of Hong Kong
{hjw0330, huding}@mail.ustc.edu.cn, minming.li@cityu.edu.hk

## Abstract

Deep learning models often struggle with distribution shifts between training and deployment environments. Distributionally Robust Optimization (DRO) offers a promising framework by optimizing worst-case performance over a set of candidate distributions, referred to as the *uncertainty set*. However, the efficacy of DRO heavily depends on the design of the uncertainty set, and existing methods often perform suboptimally due to an inappropriate or inflexible uncertainty set. In this work, we first propose a novel perspective that casts entropy-regularized Wasserstein DRO as a dynamic process of distributional exploration and semantic alignment, both driven by optimal transport (OT). This unified viewpoint yields two key new techniques: *semantic calibration*, which bootstraps semantically meaningful transport costs via inverse OT, and *adaptive refinement*, which adjusts uncertainty set using OT-driven feedback. Together, these components form an exploration-and-feedback system, where the transport costs and uncertainty set evolve jointly during training, enabling the model to better adapt to potential distribution shifts. Moreover, we provide an in-depth analysis of this adaptive process and prove theoretical guarantees of convergence. Finally, we present our experimental results across diverse distribution shift scenarios, which demonstrate that our approach significantly outperforms existing methods, achieving state-of-the-art robustness.

## 1 Introduction

Distribution shifts between training and deployment environments present a fundamental challenge to machine learning systems, often leading to significant performance degradation in real-world applications. This challenge has motivated robust learning algorithms that ensure performance under distributional uncertainty. Among these approaches, *distributionally robust optimization (DRO)* [22, 46] has emerged as a popular framework that optimizes for worst-case performance across a set of potential test distributions, known as the "*uncertainty set*". Unlike empirical risk minimization (ERM), which directly minimizes the expected loss $\ell(\theta, x)$ over the empirical training distribution $P_{\mathrm{tr}}$, DRO seeks robustness by solving the minimax problem: $\min_\theta \sup_{Q \in \mathcal{U}(P_{\mathrm{tr}})} \mathbb{E}_Q[\ell(\theta, x)]$. Here, $\mathcal{U}(P_{\mathrm{tr}})$ represents the uncertainty set that encompasses distributions in the neighborhood of $P_{\mathrm{tr}}$, capturing a range of plausible test-time deviations.

A critical challenge in implementing effective DRO lies in designing an appropriate uncertainty set. Ideally, the uncertainty set should accurately capture potential distributional shifts while remaining amenable to efficient optimization. **Optimal transport (OT)** [61] provides an elegant mathematical framework for this purpose, offering both strong theoretical foundations and intuitive physical interpretations (see Sec. 2.1). These properties make OT distances suitable for constructing meaningful

---

*Corresponding author.

39th Conference on Neural Information Processing Systems (NeurIPS 2025).

uncertainty set in distribution space. In this work, we leverage the *entropy-regularized OT distance* (also known as, *Sinkhorn distance* [16], and the formal definition is shown in Def. 2) to define our uncertainty set. Let $\mathcal{D}_\epsilon$ denote the Sinkhorn distance and the DRO problem [62, 3] is formalized as:

$$\min_\theta \sup_{Q \in \mathcal{U}(P_{\mathrm{tr}}, \delta)} \mathbb{E}_Q[\ell(\theta, x)], \quad \text{where } \mathcal{U}(P_{\mathrm{tr}}, \delta) = \{Q \in \mathcal{P}(\mathcal{X} \times \mathcal{Y}) \mid \mathcal{D}_\epsilon(P_{\mathrm{tr}}, Q) \le \delta\}. \quad (1)$$

Here $\delta > 0$ serves as the radius to control the size of allowable distributional shift; $\mathcal{P}(\mathcal{X} \times \mathcal{Y})$ denotes the set of Borel probability distributions over the data-label space "$\mathcal{X} \times \mathcal{Y}$". The uncertainty set reflects prior knowledge about potential distributional shifts. As we will demonstrate in Sec. 2.2, the Sinkhorn distance $\mathcal{D}_\epsilon(P_{\mathrm{tr}}, Q)$ introduces a reference distribution $\nu$ that guides the uncertainty set's shape. This reference distribution determines the "closeness" between distributions and the training data $P_{\mathrm{tr}}$, thereby serving as a prior knowledge about the "geometry" of the uncertainty set.

Despite its popularity, OT-based DRO (*e.g.,* (1)) still faces two major practical limitations [42]. *First*, the uncertainty set can be overly conservative, encompassing unrealistic worst-case distributions that harm practical performance. *Second*, conventional transport cost is not very effective for encoding semantic information, because the input training data is often in lack of explicit expression for semantic similarities in some practical areas such as computer vision [30]. For instance, pixel-wise distance is a poor proxy for image-level similarity [42]. As mentioned in above, the Sinkhorn distance introduce a reference distribution $\nu$, which can mitigate those issues to some extent. In particular, the distribution $\nu$ regularizes the transport plan and implicitly shapes the geometry of the uncertainty set $\mathcal{U}(P_{\mathrm{tr}}, \delta)$ by controlling where and how probability mass can be transported [62]. However, constructing an appropriate reference distribution $\nu$ remains challenging. When training data is limited, $\nu$ is typically synthesized through heuristics (e.g., mixup [67] or noise injection [26]). But, this operations may introduce unrealistic distributions in $\mathcal{U}(P_{\mathrm{tr}}, \delta)$. Moreover, we are also confronted with a tricky dilemma for constructing the uncertainty set $\mathcal{U}(P_{\mathrm{tr}}, \delta)$:

*Expanding the uncertainty set to capture meaningful shifts risks incorporating unrealistic perturbations, while overly restricting it may exclude plausible distributional variations.*

Our core idea is to design an adaptive DRO framework in which both the *semantic distances* and *uncertainty set* evolve during training, yielding a **dynamic uncertainty set**. While introducing this *flexibility* to the uncertainty set $\mathcal{U}(P_{\mathrm{tr}}, \delta)$ is appealing, it also presents significant challenges for implementation and analysis. Our proposed framework is grounded in optimal transport, offering a principled geometric interpretation that guides the algorithm design and analysis. To our best knowledge, our proposed framework is fundamentally different with existing DRO methods, from both the formulation-level and algorithmic-design level (most of them rely on static uncertainty set). We name our approach "**AdaDRO**" and summarize our contributions as follows,

- We introduce a novel theoretical perspective that frames entropy-regularized DRO as a dynamic exploration-and-alignment process driven by OT. This perspective gives rise to two core components: *semantic calibration*, which bootstraps transport costs via inverse OT, and *adaptive refinement*, which filters the uncertainty set using OT-based feedback.

- We establish theoretical convergence guarantees despite the evolving nature of the transport costs and uncertainty set, ensuring robust optimization stability throughout training. Furthermore, the OT-based structure of (1) enables seamless integration of semantic calibration and refinement into the training process.

- Extensive experiments demonstrate that our method improves both robustness and accuracy across a range of distribution shift scenarios, thereby broadening the applicability of DRO.

To illustrate the benefit of our approach, consider the 2D binary classification task shown in Figure 1. Due to limited training data, the learned decision boundary significantly deviates from the ideal boundary. The reference distribution $\nu$, constructed from Gaussian distributions around training points, may include samples that fall outside the true data manifold. Such low-quality samples can corrupt the uncertainty set and degrade classifier quality. By identifying and filtering out such samples, we obtain a boundary (blue curve) that better reflects the structure of the data distribution and more closely approximates the true decision boundary.

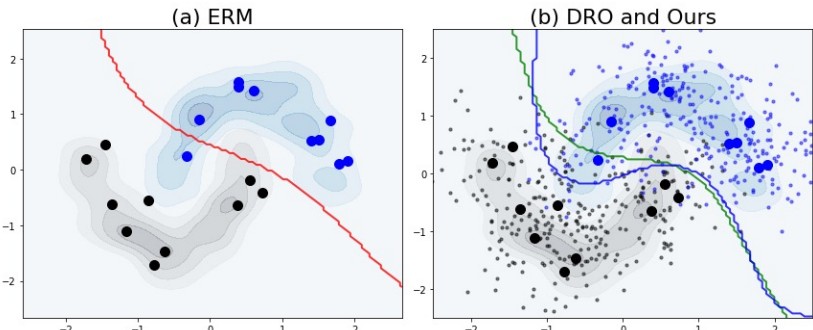

Figure 1: Binary classification on a "double moon" dataset. (a) SVM (RBF-kernel) decision boundary (red) trained on 20 samples (10 per class). (b) Comparison of Sinkhorn DRO (green) using Gaussian reference $\nu$ (small dots show 600 samples) versus our adaptive DRO (blue). Our method better captures the true data manifold by filtering low-confidence regions in $\nu$.

## 1.1 Other Related Works

Distributionally robust optimization (DRO) methods seek model parameters that minimize worst-case expected loss over an uncertainty set of distributions around the training data. Early work in this area used $\phi$-divergence uncertainty set, yielding tractable reformulations via duality; for instance, KL-DRO optimizes against distributions within a Kullback–Leibler ball around the empirical measure [29, 46]. Wasserstein DRO (WDRO) instead constructs uncertainty set based on optimal transport (OT) costs, providing geometric robustness guarantees; seminal results include finite-sample bounds and dual formulations by Blanchet and Murthy [9, 10] and extensions to high-dimensional settings by Gao and Kleywegt [23]. Entropy-regularized OT, or Sinkhorn DRO, introduces a reference measure and entropy penalty to yield smoother uncertainty set and more efficient algorithms [64, 62]. The full discussion on related works is deferred to Appendix A.1.

## 2 Preliminaries

Consider the input-output space $\Xi = \mathcal{X} \times \mathcal{Y}$, where $\mathcal{X} \subseteq \mathbb{R}^d$ is the data space and $\mathcal{Y} := [K] = \{1, \cdots, K\}$ denotes the label space containing $K$ distinct classes. Let $\mathcal{P}(\Xi)$ represent the set of Borel probability distributions supported on $\Xi$. For $1 \leq i \leq n$, each sample $p_i = (x_i, y_i)$ is drawn from an underlying distribution $\mathbb{P} \in \mathcal{P}(\Xi)$. The empirical distribution $\mathbb{P}_n = \frac{1}{n} \sum_{i=1}^{n} \delta_{p_i}$ is induced by the dataset $\{p_1, \ldots, p_n\}$, where $\delta_{p_i}$ is the Dirac measure concentrated at $p_i$. We endow $\Xi$ with a transport cost function: $C(p_i, p_j) = C^{\mathcal{X}}(x_i, x_j) + C^{\mathcal{Y}}(y_i, y_j)$, where $C^{\mathcal{X}} : \mathcal{X} \times \mathcal{X} \to \mathbb{R}$ and $C^{\mathcal{Y}} : \mathcal{Y} \times \mathcal{Y} \to \mathbb{R}$ quantify the transport costs between features and labels, respectively. This transport cost provides the foundation for defining the optimal transportation distance in (4).

We define our neural network as $f_\Theta(\cdot) = h_W \circ g_\theta(\cdot)$, where $g_\theta(\cdot) : \mathcal{X} \to \mathcal{Z}$ serves as the *encoder*, parameterized by $\theta$, which maps input data $x \in \mathcal{X}$ into a latent representation space $\mathcal{Z} \subset \mathbb{R}^{d_e}$. The function $h_W(\cdot) : \mathcal{Z} \to \mathcal{Y}$ acts as a *linear classification head* that generates predictions in the output space $\mathcal{Y}$. The classifier weight matrix $W = [w_1, \ldots, w_K] \in \mathbb{R}^{K \times d_e}$, where each $w_i$ is the class-specific weight vector for the $i$-th class. The complete set of model parameters is $\Theta = \{\theta, W\}$. For notational simplicity, we sometimes abbreviate the notation as $f(\cdot) = h \circ g(\cdot)$ and denote the classification loss function as $\ell(\cdot)$.

### 2.1 OT and Inverse-OT

Optimal Transport (OT) and Inverse Optimal Transport (IOT) offer complementary perspectives: OT seeks optimal couplings between distributions given a cost function, while IOT infers appropriate cost functions that best explains observed coupling patterns.

Let $P$ and $Q$ be probability distributions, and $\Pi(P, Q)$ denote the set of all joint distributions (called *transport plans* or *couplings*) with marginals $P$ and $Q$. Formally, any $\gamma \in \Pi(P, Q)$ satisfies $\int \gamma(p, q) \mathrm{d}q = P(p)$ and $\int \gamma(p, q) \mathrm{d}p = Q(q)$. Let $\mu$ and $\nu$ be reference measures such that $P \ll \mu$ and

$Q \ll \nu$ (i.e., $P$ and $Q$ are absolutely continuous with respect to $\mu$ and $\nu$, respectively). We begin with the entropy regularization.

**Definition 1** (Relative Entropy of Transport Plans). For any transport plan $\gamma \in \Pi(P, Q)$, the relative entropy of $\gamma$ w.r.t. product measure $\mu \otimes \nu$ is:

$$H(\gamma \mid \mu \otimes \nu) = \int \log \left( \frac{\mathrm{d}\gamma}{\mathrm{d}(\mu \otimes \nu)}(p, q) \right) \mathrm{d}\gamma(p, q). \tag{2}$$

This entropy term requires $\mathrm{supp}(P) \subseteq \mathrm{supp}(\mu)$ and $\mathrm{supp}(Q) \subseteq \mathrm{supp}(\nu)$ to remain finite. A common choice [40] uses uniform reference measures ($\mathrm{d}\mu(p)\mathrm{d}\nu(q) = \mathrm{const}$), simplifying the entropy to:

$$H(\gamma) = \mathbb{E}_{(p,q)\sim\gamma} \left[ \log(\mathrm{d}\gamma(p, q)) - 1 \right], \tag{3}$$

**Entropy-Regularized Optimal Transport**     Cuturi [16] introduces an entropic penalty to the classical OT problem, encouraging smoother couplings between distributions while minimizing transport cost. This leads to the formulation of the Sinkhorn distance:

**Definition 2** (Sinkhorn Distance (*a.k.a.* Entropy-regularized OT Distance)). Given distributions $P, Q$, a cost function $C : \Xi \times \Xi \to \mathbb{R}_+$, and regularization strength $\epsilon > 0$, the Sinkhorn distance is:

$$D_\epsilon(P, Q) = \inf_{\gamma \in \Pi(P,Q)} \left\{ \mathbb{E}_{(p,q)\sim\gamma}[C(p, q)] + \epsilon H(\gamma | \mu \otimes \nu) \right\}. \tag{4}$$

**Inverse Optimal Transport (IOT)**     IOT [57, 39, 13, 12] solves the inverse problem: inferring cost function that explains observed couplings. To capture this, we add the subscript $\theta$ to the previously defined notations "$\gamma$" and "$C$", i.e., $\gamma_\theta$ and $C_\theta$, which denote the corresponding optimal transport plan and cost learned by the model parameterized by $\theta$. In addition, we let $\bar{\gamma} \in \Pi(P, Q)$ denote an observed coupling that reflects ground-truth semantic relationships (details in Sec.3.2). Then we have the following formulation for IOT:

$$\min_\theta \mathrm{KL}(\bar{\gamma}|\gamma_\theta), \quad \text{s.t. } \gamma_\theta = \arg\min_{\gamma \in \Pi(P,Q)} \left\{ \mathbb{E}_{(p,q)\sim\gamma}[C_\theta(p, q)] + \epsilon H(\gamma) \right\}. \tag{5}$$

Here, the KL divergence $\mathrm{KL}(\bar{\gamma}|\gamma_\theta)$ quantifies how well $C_\theta$ explains the observed matching pattern $\bar{\gamma}$. The entropy term $H(\gamma)$ is defined as in (3). This bilevel structure allows us to learn transport costs $C_\theta$ such that the resulting optimal transport plan aligns with observed empirical semantic couplings.

## 2.2  Sinkhorn DRO

We now analyze the Sinkhorn DRO formulation in (1), highlighting its theoretical foundations and advantages over alternative approaches. Sinkhorn DRO extends Wasserstein DRO (WDRO) [9, 10, 23] through entropy regularization. When the transport cost $C$ is a metric and $\epsilon = 0$, the formulation (1) reduces to standard WDRO. However, WDRO typically yields discrete, adversarial worst-case distribution, making it well-suited for adversarial robustness but less effective for modeling natural distribution shifts, which often appear as continuous, structured deformations [19, 42, 56].

The introduction of entropy regularization in $D_\epsilon$ brings benefits through two complementary mechanisms: *Reference-guided uncertainty set:* The relative entropy term $H(\gamma | \mu \otimes \nu)$ leverages reference distributions $\mu$ and $\nu$ to shape the transport plan, guiding it toward more plausible distribution shifts rather than arbitrary worst-case scenarios; *Geometric Awareness:* In contrast to divergence-based DRO (e.g., KL-DRO [29]), Sinkhorn DRO incorporates transport costs $C(p, q)$ that reflects pairwise sample similarities, preserving crucial geometric information within the data space.

Although the Sinkhorn distance in (4) depends on two reference measures $\mu$ and $\nu$, following Wang et al. [62], we set $\mu = P_{\mathrm{tr}}$, the empirical training distribution. This choice is both practical and theoretically sound, as the relative entropy term differs only by a constant when $P_{\mathrm{tr}} \ll \mu$, leaving the optimization unaffected. In contrast, the reference measure $\nu$ plays a critical role: it defines the support of feasible worst-case distributions by enforcing $Q \ll \nu$. Through dual analysis, Wang et al. [62] derived the closed-form expression for the worst-case distribution. When the optimal Lagrange multiplier $\lambda > 0$, the worst-case distribution $Q^\lambda$ has the density:

$$dQ^\lambda(q) = \mathbb{E}_{p\sim P_{\mathrm{tr}}} \left[ \frac{e^{(\ell(q)-\lambda C(p,q))/(\lambda\epsilon)}}{\mathbb{E}_{u\sim\nu} \left[ e^{(\ell(u)-\lambda C(p,u))/(\lambda\epsilon)} \right]} \right] d\nu(q). \tag{6}$$

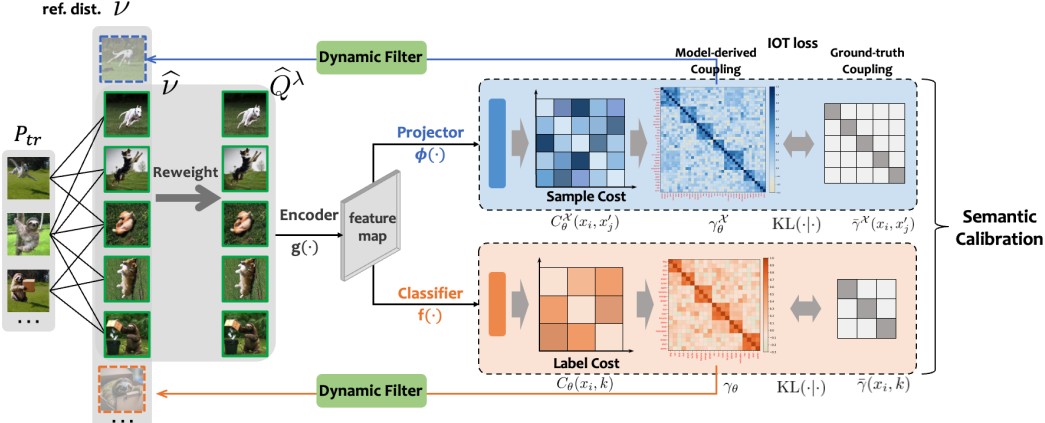

Figure 2: Overview of the AdaDRO framework. Starting from the empirical distribution $P_{\mathrm{tr}}$, we construct a reference distribution $\nu$ and dynamically filter out low-confidence samples to obtain the refined set $\widehat{\nu}$. Semantic calibration is performed by aligning model-derived transport couplings with ground-truth couplings using inverse optimal transport (IOT) losses. This guides the construction of semantically meaningful and adaptive uncertainty sets.

**The role of $\nu$.** The formula (6) shows that the worst-case distribution $Q^\lambda$ is essentially a reweighted version of $\nu$, with weights determined by the loss function $\ell$ and transport cost $C$. The absolute continuity condition $Q \ll \nu$ ensures that the support of $Q^\lambda$ remains within that of $\nu$, highlighting why the design of $\nu$ is crucial for effective distributional robustness. By dynamically refining the reference distribution $\nu$, we can induce a *dynamic uncertainty set* that evolves with the learning dynamic to better reflect the underlying data geometry, which has not been explored previously.

## 3 Our Method: AdaDRO

To address the dilemma highlighted in Sec. 1, we propose a self-adaptive framework that jointly learns semantic costs and refines uncertainty set based on OT.

**High-level idea** We reformulate the Sinkhorn DRO objective as a bilevel optimization whose upper-level and lower-level objectives are OT and IOT objectives, respectively (Sec. 3.1). This perspective inspires us to incorporate a more well-designed IOT for semantic calibration in Sec. 3.2; moreover, the geometric implication of OT coupling can serve as a feedback to refine the uncertainty set (Sec. 3.3). The overall framework is illustrated in Figure 1.

### 3.1 Sinkhorn DRO Revisited: A New Perspective from OT

In this section, we introduce a new perspective for Sinkhorn DRO from OT. First, we cast Sinkhorn DRO as a *semi-relaxed optimal transport (OT) problem*, where the benefit is that the worst-case distribution $Q^\lambda$ (defined in (6)) emerges naturally from the optimal transport plan. Additionally, we reinterpret the softmax cross-entropy loss as a form of *inverse OT*. Taken together, these insights enable the entire learning dynamics of DRO to be formulated through the OT/IOT objectives, so that it paves the way for building our framework and theoretical analysis in the following subsections.

**Distribution Exploration via OT.** The optimal coupling of OT in Def. 2 represents the most efficient way to transport a source distribution $P$ to the target distribution $Q$. By relaxing the target marginal constraint, *i.e.,* replace the "$\gamma \in \Pi(P, Q)$" with "$\gamma \in \Pi(P) := \left\{ \gamma \mid \int \gamma(p, q) \mathrm{d}q = P(p) \right\}$",

$$D_\epsilon(P, \star) = \inf_{\gamma \in \Pi(P)} \left\{ \mathbb{E}_{(p,q) \sim \gamma}[C(p, q)] + \epsilon H(\gamma \mid P \otimes \nu) \right\}. \tag{7}$$

This relaxation considers couplings that preserve only the source marginal, allowing the distribution $P$ to "explore" the target space. The exploration is guided by the transport cost function $C$ and

the reference measure $\nu$. Under an appropriately chosen cost, this exploration process recovers the Sinkhorn DRO worst-case distribution $Q^\lambda$ as its target marginal.

**Lemma 3.1.** *Consider the DRO objective in* (1) *with $P_{\mathrm{tr}}$ and $\nu$ being the source and reference measures. If the transport cost in the semi-relaxed OT problem* (7) *is taken as $C'(p,q) = \frac{1}{\lambda}\left(\lambda C(p,q) - \ell(q)\right)$, then the optimal transport plan $\gamma^* \in \Pi(P_{\mathrm{tr}})$ has its target marginal equal to the worst-case distribution $Q^\lambda$ in* (6).

**Classification as Inverse OT.** Complementing the above view, we interpret classification through the lens of inverse OT. In this formulation, the goal is to learn a transport cost (implicitly parameterized by the model) such that the resulting optimal coupling aligns with the known label structure. To make this concrete, we define the ground-truth coupling and learned cost as:

$$\bar{\gamma}(x_i, k) := Q^\lambda(x_i)\delta_{y_i=k}, \qquad C_\theta(x_i, k) := c - \langle g_\theta(x_i), w_k \rangle, \tag{8}$$

where $\bar{\gamma}(x_i, k)$ encodes the coupling between sample $x_i$ and its true label $y_i$, and $C_\theta(x_i, k)$ defines the model-learned transport cost to class $k \in [K]$.

As shown by Shi et al. [53], when using the cost $C_\theta$ in the semi-relaxed OT formulation (7), the resulting optimal plan $\gamma_\theta$ takes the form of a softmax distribution: $\gamma_\theta(x_i, k) = \frac{\exp(\langle g(x_i), w_k \rangle/\epsilon)}{\sum_{j=1}^{K} \exp(\langle g(x_i), w_j \rangle/\epsilon)}$, and the inverse OT objective, defined as $\mathrm{KL}(\bar{\gamma}|\gamma_\theta)$, becomes:

$$\mathrm{KL}\left(\bar{\gamma}|\gamma_\theta\right) = -\mathbb{E}_{x_i \sim Q^\lambda} \log \frac{\exp\left(\langle g(x_i), w_{y_i} \rangle/\epsilon\right)}{\sum_{k=1}^{K} \exp(\langle g(x_i), w_k \rangle/\epsilon)} + \mathrm{Const}. \tag{9}$$

This expression exactly recovers the temperature-scaled cross-entropy loss commonly used in classification tasks.

**Remark 1.** This analysis shows that the DRO objective in (1) can be cast as a bilevel optimization [32] grounded in OT: **Lower level (Exploration):** exploring the worst-case distribution $Q^\lambda(\theta)$ by a semi-relaxed OT (as Lemma 3.1). **Upper level (Alignment):** updating parameters $\theta$ by solving an inverse OT problem that aligns the model's induced transport plan with ground-truth semantic labels. This OT-based perspective not only provides theoretical clarity but also suggests natural mechanisms for adaptive algorithmic design, as explored in the following subsections.

### 3.2 Semantic Calibration via Inverse OT

Recall from Sec. 2 that we defined our overall transport cost as the sum of feature and label components: $C(p_i, p_j) = C^{\mathcal{X}}(x_i, x_j) + C^{\mathcal{Y}}(y_i, y_j)$. The effectiveness of DRO critically depends on how well these costs capture semantic relationships in the data. Rather than manually specifying these costs, we propose a principled approach to learn them directly from the data using IOT.

**Calibrating the Sample-level Transport Cost $C^{\mathcal{X}}$.** To ensure that transport costs capture task-relevant semantic relationships, we propose learning $C_\theta^{\mathcal{X}}$ through a principled IOT formulation. We define the parameterized transport cost as:

$$C_\theta^{\mathcal{X}}(x_i, x_j) = c - \mathrm{Ker}\left(g_\theta(x_i), g_\theta(x_j)\right), \tag{10}$$

where $c$ is a constant ensuring non-negativity, $g_\theta(x)$ represents the embedding of sample $x$, and $\mathrm{Ker}(\cdot, \cdot)$ is a kernel function (e.g., cosine or Gaussian) measuring similarity in the embedding space.

Given a sample $x_i$ drawn from the worst-case distribution $Q^\lambda$, we generate a semantically equivalent counterpart $x_i' \sim \mathrm{Aug}(x_i)$ using standard data augmentation techniques (e.g., cropping, rotation). These pairs $(x_i, x_i')$ define ground-truth semantic matching encoded as a coupling:

$$\bar{\gamma}^{\mathcal{X}}(x_i, x_j') = \mathbb{I}[j = i], \tag{11}$$

where $\mathbb{I}[\cdot]$ is the indicator function. This coupling represents the semantically meaningful matches that our transport cost should respect.

To calibrate $C_\theta^{\mathcal{X}}$, we formulate the following IOT problem:

$$\min_\theta \mathrm{KL}(\bar{\gamma}^{\mathcal{X}}|\gamma_\theta^{\mathcal{X}}), \quad \text{s.t. } \gamma_\theta^{\mathcal{X}} = \arg \min_{\gamma \in \Pi(Q^\lambda)} \mathbb{E}_{(x_i, x_j') \sim \gamma} [C_\theta^{\mathcal{X}}(x_i, x_j')] + \epsilon H(\gamma). \tag{12}$$

Similar to (7), $\Pi(Q^\lambda)$ denotes the set of couplings whose first marginal is $Q^\lambda$, *i.e.,* $\Pi(Q^\lambda) = \{\gamma \mid \sum_j \gamma(x_i, x_j') = Q^\lambda(x_i)\}$. This optimization aims to learn parameters $\theta$ such that the transport plan $\gamma_\theta^{\mathcal{X}}$, induced by the cost function $C_\theta^{\mathcal{X}}$, aligns with the ground-truth coupling $\bar{\gamma}^{\mathcal{X}}$ defined in (11).

**Calibrating Label-level Transport Cost $C^{\mathcal{Y}}$.** To define meaningful transport costs between class labels, we need to quantify semantic relationships among them. Unlike data points, class labels lack explicit representations in the embedding space. Fortunately, the "Classification as Inverse OT" formulation from Sec. 3.1 offers a principled mechanism for semantic calibration in label space. To formalize this connection, we revisit the classification loss through the lens of inverse OT:

$$\min_{\theta} \mathrm{KL}\left(\bar{\gamma}|\gamma_{\theta}\right), \quad \text{s.t.} \quad \gamma_{\theta} = \arg \min_{\gamma \in \Pi(Q^{\lambda})} \left\{ \mathbb{E}_{(q_i, k) \in \gamma} [C_{\theta}(q_i, k)] + \epsilon H(\gamma) \right\}. \tag{13}$$

Here, the cost function is defined as $C_{\theta}(q_i, k) = c - \langle g_{\theta}(x_i), w_k \rangle$, which can be seen as a special case of a kernel-based cost: $C_{\theta}(q_i, k) = c - \mathrm{Ker}(g_{\theta}(x_i), w_k)$, where $\mathrm{Ker}(\cdot, \cdot)$ is a linear kernel. A fundamental property of such kernels, known as the reproducing property [47], establishes that: $\langle \mathrm{Ker}(w_k, \cdot), \mathrm{Ker}(w_{k'}, \cdot) \rangle = \mathrm{Ker}(w_k, w_{k'})$. This property reveals that standard cross-entropy training, which is equivalent to minimizing an inverse OT objective as shown in Sec. 3.1, implicitly calibrates the semantic structure in the label space. Leveraging this insight, we define the label-level transport cost as:

$$C_{\theta}^{\mathcal{Y}}(k, k') = c - \mathrm{Ker}(w_k, w_{k'}), \quad \forall k, k' \in [K] \tag{14}$$

where $c$ is a constant ensuring non-negativity and $\mathrm{Ker}(\cdot, \cdot)$ is the linear kernel. Classes with similar weight vectors will have lower transport costs between them, reflecting their semantic proximity. A similar idea of using classifier weight vectors $w_k$ to encode label semantics has also appeared in prior work, such as [58], although they do not interpret this mechanism by inverse OT alignment.

### 3.3 Our Overall Algorithm

As discussed in Sec. 1, the reference distribution $\nu$ serves as a crucial prior that defines the support of the DRO uncertainty set. However, high-quality priors are rarely available: an overly broad $\nu$ may include implausible samples, while an overly restrictive one weakens robustness guarantees. The OT formulations in Sec.3.1–3.2 incorporate relaxed marginal constraints, reducing the need to forcefully match noisy or outlier samples [5, 14]. This flexibility enables the transport coupling to reflect the intrinsic structure of the data [49, 64]. Motivated by this observation, we introduce an OT-based feedback mechanism that adaptively filters the reference distribution $\nu$, thereby refining the uncertainty set during training.

**Refining $\nu$ via OT Feedback.** We have introduced two Inverse OT frameworks for semantic calibration—feature-level IOT in (12) and label-level IOT in (13). A key question now arises: how can we leverage the resulting couplings to dynamically refine the uncertainty set?

To address this, we propose a confidence-based filtering mechanism that evaluates the alignment strength between source-target pairs based on the inferred coupling. We illustrate this using the feature-level IOT framework in (12). The key idea is to compare the model-induced coupling $\gamma_{\theta}$ with the ground-truth coupling $\bar{\gamma}^{\mathcal{X}}$ (defined in (11)). For any matched pair $(p, q) \sim \bar{\gamma}^{\mathcal{X}}$, we define its *matching confidence* as $\gamma_{\theta}^{\mathcal{X}}(p, q)$, which quantifies how strongly the model aligns $p$ with $q$. A global average confidence is then computed as $\tau_1 = \mathbb{E}_{(p,q) \sim \bar{\gamma}^{\mathcal{X}}} \gamma_{\theta}^{\mathcal{X}}(p, q)$.

Because some targets may inherently be more difficult to match, particularly when the marginal distribution of the ground-truth coupling $\bar{\gamma}^{\mathcal{X}}$ is imbalanced, we define a target-specific *local matching capacity* for each $q$: $\tau_2(q) = \mathbb{E}_p \gamma_{\theta}^{\mathcal{X}}(p, q)$, which quantifies how much mass is transported to $q$ across all source. A larger $\tau_2(q)$ indicates that the target $q$ is easier to match. Combining global and local signals, we define a dynamic threshold:

$$\tau(q) = \tau_1 \cdot \frac{\tau_2(q)}{\max_{q'} \tau_2(q')}. \tag{15}$$

Each source $p$ is retained in the filtered reference distribution $\widehat{\nu}$ only if it aligns well with at least one target $q$:

$$\widehat{\nu}(p) = \frac{\nu(p) \cdot \mathbb{I}[\exists q : \bar{\gamma}^{\mathcal{X}}(p,q) > 0 \wedge \gamma_{\theta}^{\mathcal{X}}(p,q) \geq \tau(q)]}{\chi}, \tag{16}$$

where $\chi$ normalizes $\widehat{\nu}$ to form a valid distribution.

This filtering mechanism offers several advantages: **(1)** It establishes a feedback loop between model training dynamics and uncertainty set refinement, enabling adaptive robust optimization; **(2)** It uses

---

**Algorithm 1** OT Driven Adaptive Distributionally Robust Optimization (AdaDRO)

---

**Require:** Initial parameter $\theta_0$, step size $\alpha > 0$, total iterations T.
 1: Train $\theta$ by the joint objective in (12) and (13) for $N_0$ epochs. ▷ Semantic calibrate by IOT
 2: **for** $t = 0, 1, \ldots, T - 1$ **do**
 3:     $\widehat{\nu}_t \leftarrow \text{FilterReference}(\nu, \theta_t)$    ▷ Filter via (16)
 4:     $\widehat{Q}_t^\lambda \leftarrow \text{SinkhornWorstCase}(P_{\text{tr}}, \widehat{\nu}_t, C_{\theta_t}, \lambda)$ ▷ Compute worst-case distribution via (6).
 5:     Estimate gradient $\texttt{grad}_t$ via RT-MLMC estimation for the IOT objectives (13) and (12) ▷ See Algorithm 2 in Appendix B.4
 6:     Update $\theta_{t+1} \leftarrow \theta_t - \alpha_t \texttt{grad}_t$
 7: **end for**

---

sample-specific thresholds to account for varying target difficulties, overcoming a key limitation of static filtering rules. This design of our filtering mechanism is guided by the geometric intuition behind transport couplings, as previously discussed. We also draw conceptual connections to the well-known thresholding strategy introduced in FreeMatch [63], which adaptively filters samples in semi-supervised learning by combining *global* and *local* (class-specific) thresholds. Such methods eliminate the reliance on fixed, manually tuned thresholds and leverage data-driven signals to inform selection.

**Overall Algorithm.** We apply the filtering rule in both semantic calibration tasks—feature-level IOT in (12) and label-level IOT in (13)—to obtain a refined reference distribution $\widehat{\nu}$. We then perform Sinkhorn distributionally robust optimization using $\widehat{\nu}$. The full procedure of our OT-driven adaptive robust algorithm is summarized in Algorithm 1, and its schematic overview is provided in Figure 2.

**Theorem 3.2** (Convergence of AdaDRO). *Let the objective at iteration $t$ to be defined as $F_t(\theta) = \mathbb{E}_{q \sim \widehat{Q}_t^\lambda}[\ell(\theta, q)]$. Under mild smoothness assumptions, Algorithm 1 with decaying step size $\alpha_t = \alpha/\sqrt{t+1}$ converges to a stationary point after at most $T = O\left(\epsilon^{-4} \log \frac{1}{\epsilon}\right)$ iterations:*

$$\min_{0 \le t < T} \mathbb{E}\left[\|\nabla F_t(\theta_t)\|^2\right] \le \epsilon^2. \tag{17}$$

*Here the total sample complexity $O\left(\epsilon^{-4} \log^2 \frac{1}{\epsilon}\right)$.*

Theorem 3.2 establishes the convergence of our adaptive Sinkhorn DRO algorithm to a stationary point, despite two major challenges: **(i)** the **dynamically evolving uncertainty set** $\widehat{Q}_t^\lambda$ that depends on both the model and the filtered reference distribution, and **(ii)** the presence of **nested expectations** in the objective, which renders unbiased gradient estimation intractable.

To handle **(i)**, we ensure that both the transport cost $C_\theta$ and the filtered reference distribution $\widehat{\nu}_\theta$ vary smoothly with $\theta$, as shown in Proposition B.1. This ensures that changes to the uncertainty set remain controlled during training.

For challenge **(ii)**, the objective $F_t(\theta) = \mathbb{E}_{q \sim \widehat{Q}_t^\lambda}[\ell(\theta, q)]$ involves a composition of expectations. Because the worst-case distribution $\widehat{Q}_t^\lambda(q)$, as we introduced in (6), is given by

$$\widehat{Q}_t^\lambda(q) = \phi(q)\widehat{\nu}_t(q), \quad \text{and} \quad \phi(q) := \mathbb{E}_{p \sim P_{\text{tr}}}\left[\frac{e^{(\ell(q) - \lambda C(p,q))/(\lambda\epsilon)}}{\mathbb{E}_{u \sim \widehat{\nu}_t}\left[e^{(\ell(u) - \lambda C(p,u))/(\lambda\epsilon)}\right]}\right]. \tag{18}$$

Here the denominator of $\phi(q)$ is also an expectation over $u \sim \widehat{\nu}_t$. This *nested structure* makes it impossible to construct an unbiased gradient estimator using standard Monte Carlo, and naive stochastic approximations often incur intractable cost [38, 62]. To address this, we adopt the multilevel Monte Carlo with Randomized Truncation (MLMC-RT) technique [28], a method has been used in robust optimization to trade off bias, variance, and computational cost efficiently. Related works such as [38] use MLMC for estimating gradients in worst-case distributional settings. In our case, we adapt MLMC-RT to the setting of *dynamically changing uncertainty set* and show that it remains theoretically sound.

Together, these properties enable our AdaDRO to converge to an $\epsilon$-stationary point using a total of $O(\epsilon^{-4} \log^2 \frac{1}{\epsilon})$ samples (as in Theorem 3.2)—only logarithmically worse than the known lower bound for general non-convex stochastic optimization [1]. This result demonstrates that *robust optimization with dynamically adaptive uncertainty sets and semantic costs can also be performed efficiently at scale*. Our full theoretical details are provided in Appendix B.

# 4 Experiments

We evaluate the proposed AdaDRO framework across a diverse set of benchmarks to assess its effectiveness under various types of distribution shifts.

## 4.1 Experimental Setup

**Datasets.** We evaluate on three widely studied distribution shift settings: *Colored MNIST* [2], which tests robustness under spurious correlations; *Waterbirds* [50], a real-world dataset with strong background-label correlation; *CelebA* [43], a benchmark for facial attribute recognition. We also evaluate on several long-tailed benchmark datasets: *CIFAR-10-LT* and *CIFAR-100-LT* [37].

**Baselines.** We compare AdaDRO with the following methods: **KL-DRO** [46], Wasserstein DRO (**WDRO**) [9], Sinkhorn DRO **SDRO** [62], a unified DRO method **UniDRO** [11] , semantic-aware robust methods such as **GroupDRO** [50], an invariant risk minimization method **IRM** [2].

Our models are implemented with PyTorch on a single NVIDIA RTX 6000 Ada GPU using the AdamW optimizer [45]. For AdaDRO, we use cosine similarity as the kernel in (10), and employ basic augmentations (flip, crop) for semantic calibration in Sec. 3.2 unless otherwise specified.

## 4.2 Experimental Results

AdaDRO is broadly applicable thanks to its flexible reference distribution design. While this flexibility enables adaptation to various settings, our method still relies on prior knowledge to construct the reference distribution, which may limit its applicability in some cases. Nevertheless, domain priors can often be effectively incorporated, as in many specialized robust learning approaches. For instance, in imbalanced settings, the reference distribution $\nu$ can be constructed through reweighting or resampling. We evaluate AdaDRO across diverse scenarios, including spurious correlation, group shift, and long-tail generalization. Additional experimental details are in Appendix D.

Tables 1 and 2 show part of the results on Colored MNIST and Waterbirds. AdaDRO consistently achieves top worst-group accuracy, outperforming or matching specialized methods like GroupDRO.

Table 1: Accuracy (%) on Colored MNIST under high spurious correlation. AdaDRO achieves the highest robustness across methods.

| Method | Avg Accuracy | Worst-group Acc | Gap |
|---|---|---|---|
| IRM [2] | 79.5 | 75.3 | 4.2 |
| GroupDRO [50] | 78.4 | 73.0 | 5.4 |
| KL-DRO [46] | 74.9 | 68.5 | 6.4 |
| WDRO [9] | 75.8 | 69.1 | 6.7 |
| SDRO [62] | 79.0 | 72.6 | 6.4 |
| UniDRO [11] | 67.1 | 47.8 | 20.7 |
| AdaDRO (ours) | **87.2** | **79.9** | 7.3 |

Table 2: Accuracy (%) on Waterbirds. AdaDRO significantly improves the SDRO.

| Method | Avg Accuracy | Worst-group Acc | Gap |
|---|---|---|---|
| IRM [2] | 82.6 | 73.2 | 9.4 |
| GroupDRO [50] | 86.3 | **80.5** | 5.8 |
| KL-DRO [46] | 81.0 | 71.5 | 9.5 |
| SDRO [62] | 85.0 | 75.4 | 9.6 |
| AdaDRO (ours) | **87.1** | **80.5** | 6.6 |

Compared with other baselines, our method offers two main benefits. First, it performs a label-agnostic, sample-level semantic calibration as in (12). Second, it introduces a coupling-based filtering mechanism derived from the coupling introduced by (12) and (13). An intuitive example is the presence of noisy labels as shown in Table 3: when a portion of labels are flipped, sample-level

calibration (12) can still reliably capture semantics. This is because the process is unsupervised, similar to contrastive learning [53], and thus remains robust to label noise. Building on this, the coupling-based filtering mechanism can discards samples with wrong label, enabling the cross-entropy to train on a cleaner subset of the data. In contrast, methods that rely heavily on label supervision, such as ERM with augmentation, suffer from an inherent "garbage in, garbage out" problem. No augmentation can correct mislabeled examples, and these label errors inevitably propagate through the learning process, compromising model robustness.

A similar advantage is observed in Colored MNIST as in Table 4. Even when augmentation occasionally alters the spurious attribute (e.g., color), standard ERM still learns entangled representations, as the label remains strongly correlated with color. In contrast, our label-agnostic calibration encourages the model to identify truly invariant features—those that remain stable under augmentation—naturally prioritizing shape over color in the absence of a strong supervisory signal.

Table 3: Accuracy (%) on CIFAR-100 with 50% label noise. Half of the training labels are randomly corrupted. We report test accuracy under different regularization strengths $\lambda$.

| Accuracy (%) | $\lambda$=0.5 | $\lambda$=1.0 | $\lambda$=2.0 | $\lambda$=3.0 | $\lambda$=5.0 |
|---|---|---|---|---|---|
| AdaDRO (Ours) | $52.5 \pm 0.6$ | $54.2 \pm 0.5$ | $\mathbf{57.2 \pm 0.8}$ | $55.6 \pm 1.1$ | $53.2 \pm 0.8$ |
| SDRO [62] | $42.5 \pm 0.3$ | $41.2 \pm 0.7$ | $44.1 \pm 0.3$ | $44.8 \pm 0.5$ | $43.9 \pm 0.3$ |

Table 4: Accuracy (%) on Colored MNIST benchmark under strong spurious correlations. *ERM + Aug* denotes empirical risk minimization with carefully designed data augmentation.

| Method | ERM + Aug | KL-DRO [46] | IRM [2] | SDRO [62] | AdaDRO (Ours) |
|---|---|---|---|---|---|
| Accuracy (%) | $43.5 \pm 0.6$ | $41.2 \pm 0.5$ | $45.1 \pm 1.0$ | $44.8 \pm 0.6$ | $\mathbf{57.2 \pm 0.8}$ |

## 5 Conclusion

We proposed a novel OT-based framework for adaptive robust optimization, where transport costs and uncertainty set co-evolve through inverse optimal transport. By combining semantic calibration with OT-based feedback, our method forms a closed-loop process that adapts to distribution shifts during training. The experiments show consistent gains over strong DRO baselines, and our framework opens promising directions for enhancing the flexibility of DRO methods. Moreover, our framework offers a new perspective for enhancing the flexibility of DRO through the lens of OT, paving the way for incorporating richer OT tools into robust learning.

## Acknowledgments and Disclosure of Funding

This work was supported in part by the National Key R&D program of China through grant 2021YFA1000900, the NSFC through grants No. 62432016 and No. 62272432, and the Provincial NSF of Anhui through grant 2208085MF163. The authors would also like to thank the anonymous reviewers for their valuable comments and suggestions.

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

# A    Omitted Materials

## A.1    Other Related Works

**Metric learning.**    The inverse optimal transport problem shares conceptual and methodological similarities with metric learning, thoroughly examined by [4, 34]. Metric learning focuses on estimating pairwise distance metrics from observed interactions, typically constraining solutions to specific metric classes such as Mahalanobis distances [66, 65, 18]. However, while metric learning directly optimizes distances, inverse optimal transport (IOT) [57, 39, 13, 48] employs a more sophisticated bilevel optimization structure that accounts for global coupling relationships between point sets, significantly increasing its complexity. Cuturi and Avis [17] conceptualized the inverse OT problem as a metric learning task, an approach later applied to generative models [24]. Taking a different direction, Frogner et al. [21] proposed constructing cost functions using external priors such as word embeddings, where semantic similarity between labels is derived from Euclidean distances in pre-trained word2vec spaces (as demonstrated with the Flickr dataset). Though intuitive, this approach introduces external dependencies and may not adequately capture task-specific semantic relationships.

**Distributional robustness.**    Distributionally robust optimization (DRO) [38, 52] provides a principled framework for learning models that remain reliable under test-time distribution shifts. The central idea is to optimize model performance against the worst-case distribution within a predefined uncertainty set around the training distribution. A key ingredient of DRO is its *dual formulation*, which enables tractable reformulations and efficient algorithms [31]. This paradigm has been extended to group-wise robustness, as in Group DRO [51], which seeks to ensure balanced performance across predefined or latent subgroups—even when some groups are underrepresented. Recent methods such as [59] further refine this idea by incorporating logit adjustment to improve performance parity across groups without explicit group labels.

**Other related robustness methods.**    Another prominent class of techniques for handling distribution shift is importance weighting (IW) [36], which reweights training samples to better match the test distribution. Several recent advances have extended IW to broader settings. For instance, Shu et al. [54] proposed a meta-learning framework that learns a sample-weighting function using a small unbiased validation set, effectively addressing issues like class imbalance and label noise. Fang et al. [19] introduced an iterative method that jointly estimates weights and performs classification in an end-to-end fashion. More recently, Fang et al. [20] proposed Generalized Importance Weighting (GIW), which can handle complex shifts, including scenarios where the training and test distributions differ in support.

## A.2    Reproducing Kernel Hilbert Space (RKHS)

Reproducing Kernel Hilbert Space (RKHS) [8, 7] serves as a cornerstone in modern machine learning and functional analysis, providing a rigorous mathematical framework for kernel-based methods. This paper outlines the fundamental concepts of RKHS, presents its formal definition, and highlights its significance in computational applications.

In many areas of machine learning and statistics, the representation and manipulation of data in high-dimensional spaces are crucial for understanding complex patterns. Reproducing Kernel Hilbert Space (RKHS) provides an elegant framework that combines the computational efficiency of kernels with the theoretical rigor of Hilbert spaces. RKHS is particularly notable for its role in kernel methods, such as support vector machines and Gaussian processes.

The power of RKHS lies in its ability to map data into a potentially infinite-dimensional space while enabling computations through finite-dimensional kernel functions. We begin by defining the key components that constitute a Reproducing Kernel Hilbert Space.

**Definition 3** (Reproducing Kernel)**.** Let $\mathcal{H}$ be a Hilbert space of functions defined on a non-empty set $\mathcal{X}$. A function $\mathrm{Ker} : \mathcal{X} \times \mathcal{X} \to \mathbb{R}$ is called a *reproducing kernel* for $\mathcal{H}$ if: (i) For all $x \in \mathcal{X}$, $\mathrm{Ker}(\cdot, x) \in \mathcal{H}$. (ii) For all $f \in \mathcal{H}$ and $x \in \mathcal{X}$, the *reproducing property* holds: $f(x) = \langle f, \mathrm{Ker}(\cdot, x) \rangle_{\mathcal{H}}$.

**Definition 4** (Reproducing Kernel Hilbert Space)**.** A Hilbert space $\mathcal{H}$ of functions on $\mathcal{X}$ is called a *Reproducing Kernel Hilbert Space* if there exists a reproducing kernel $\mathrm{Ker}$ for $\mathcal{H}$.

The reproducing property ensures that the evaluation of a function $f \in \mathcal{H}$ at a point $x \in \mathcal{X}$ can be expressed as an inner product in $\mathcal{H}$. This property underpins the computational efficiency of kernel-based methods. In our setting, we leverage this property to construct semantic transport costs (see (10) and (14)) that reflect similarities in latent and label spaces, while enabling efficient gradient-based optimization.

## A.3 Differences and Connections with other DRO methods

**Differences and Connections with Wasserstein DRO and KL DRO** (1) Flexibility of the Cost Function: WDRO requires the transport cost function $C(\cdot, \cdot)$ to be a metric, whereas Sinkhorn DRO only imposes the weaker condition of non-negativity $C(x, y) \geq 0$. This makes Sinkhorn DRO more flexible. In WDRO without regularization, the worst-case distribution is typically discrete, while in Sinkhorn DRO, the worst-case distribution $Q^\star$ is absolutely continuous with respect to the reference measure $\nu$. The reference measure $\nu$ can be interpreted as prior knowledge about the structure of the worst-case distribution, which enhances both the interpretability and practical applicability of the model. Notably, as the regularization parameter $\epsilon \to 0$, the dual objective of Sinkhorn DRO converges to that of Wasserstein DRO because the Sinkhorn distance is an entropy-regularized version of the Wasserstein distance. (2) Connection to KL DRO: When the reference distribution $\nu = P_{\text{tr}}$, and the transport cost function $C(\cdot, \cdot) \equiv 0$, Sinkhorn DRO reduces to the KL-DRO problem. In this case, the entropy regularization term of the Sinkhorn distance simplifies to the KL divergence, which measures the cross-entropy difference between the worst-case distribution and the input distribution.

## A.4 Solutions for Entropy-Regularized Optimal Transport

For completeness, we summarize the closed-form solutions for entropy-regularized optimal transport (OT) problems under different levels of marginal constraint relaxation. These results are standard in the OT literature and can be found in [53]. Such relaxations provide analytical forms for the optimal transport plan $\gamma$, which significantly reduce computational complexity and enable efficient integration into modern learning algorithms.

**Remark 2.** Let $\Pi(P, Q)$ denote the set of couplings (joint probability measures) with specified marginals:

$$\Pi(P, Q) = \left\{ \gamma \mid \int \gamma(p, q) \, dq = P(p), \ \int \gamma(p, q) \, dp = Q(q) \right\}, \tag{19}$$

$$\Pi(P) = \left\{ \gamma \mid \int \gamma(p, q) \, dq = P(p) \right\}, \tag{20}$$

$$\Pi(1) = \left\{ \gamma \mid \int \gamma(p, q) \, dp dq = 1 \right\}. \tag{21}$$

**Relaxed Marginal Constraint:** $\gamma \in \Pi(P)$   This semi-relaxed formulation preserves only the source marginal. The entropy-regularized OT problem is defined as:

$$\gamma^* = \arg \min_{\gamma \in \Pi(P)} \ \mathbb{E}_{(p,q) \sim \gamma}[C(p, q)] + \epsilon H(\gamma \mid P \otimes \nu), \tag{22}$$

where $H(\gamma \mid P \otimes \nu)$ denotes the relative entropy (defined in (2)) with respect to the product measure of $P$ and a reference measure $\nu$. The optimal coupling has a closed-form expression:

$$\gamma^*(p, q) = \frac{\exp\left(-\frac{C(p,q)}{\epsilon}\right)}{\int \exp\left(-\frac{C(p,u)}{\epsilon}\right) d\nu(u)} \cdot P(p). \tag{23}$$

This expression yields a softmax-like kernel around each point $p$, shaped by the cost $C$ and regularization $\epsilon$, and normalized with respect to the reference measure $\nu$.

**Fully Relaxed Constraint:** $\gamma \in \Pi(1)$   In this most relaxed case, only the total mass is constrained to be 1:

$$\gamma^* = \arg \min_{\gamma \in \Pi(1)} \ \mathbb{E}_{(p,q) \sim \gamma}[C(p, q)] + \epsilon H(\gamma). \tag{24}$$

The optimal transport plan admits the following form:

$$\gamma^*(p, q) = \frac{\exp\left(-\frac{C(p,q)}{\epsilon}\right)}{Z}, \quad \text{where } Z = \iint \exp\left(-\frac{C(u,v)}{\epsilon}\right) dudv. \tag{25}$$

This represents a global Gibbs distribution over the joint space $\mathcal{X} \times \mathcal{Y}$ and is often used for unconstrained matching.

**Fully Constrained OT:** $\gamma \in \Pi(P, Q)$  This is the standard OT formulation with both source and target marginals enforced. In this case, no closed-form solution exists, but iterative scaling via the Sinkhorn algorithm [55, 6] yields the unique entropy-regularized solution:

$$\gamma^{(0)}(p, q) = \exp\left(-\frac{C(p,q)}{\epsilon}\right), \tag{26}$$

$$\gamma^{(k+1)}(p, q) = \frac{\gamma^{(k)}(p,q)}{\int \gamma^{(k)}(p,q')\,\mathrm{d}q'} \cdot P(p), \quad \text{(row normalization)} \tag{27}$$

$$\gamma^{(k+2)}(p, q) = \frac{\gamma^{(k+1)}(p,q)}{\int \gamma^{(k+1)}(p',q)\,\mathrm{d}p'} \cdot Q(q), \quad \text{(column normalization)} \tag{28}$$

This procedure is guaranteed to converge to the unique solution $\gamma^* \in \Pi(P, Q)$ satisfying the marginal constraints.

These closed-form and iterative solutions illustrate how different constraint relaxations influence the structure of optimal couplings. In our framework (see Sec. 3.1 and Sec. 3.2), we make use of the semi-relaxed formulation $\gamma \in \Pi(P)$ due to its analytical tractability and its suitability for dynamic uncertainty set modeling under limited supervision.

## A.5  Proof of Lemma 3.1

*Proof.* We begin by considering the semi-relaxed optimal transport (OT) problem where the coupling $\gamma$ preserves only the source marginal $P_{\mathrm{tr}}$, with reference distribution $\nu$, and a modified transport cost:

$$C'(p, q) := \frac{1}{\lambda}\left(\lambda C(p, q) - \ell(q)\right). \tag{29}$$

We seek the optimal coupling $\gamma^* \in \Pi(P_{\mathrm{tr}})$ that minimizes the entropy-regularized OT objective:

$$\gamma^* = \arg\min_{\gamma \in \Pi(P_{\mathrm{tr}})} \mathbb{E}_{(p,q)\sim\gamma}[C'(p, q)] + \epsilon H(\gamma \mid P_{\mathrm{tr}} \otimes \nu). \tag{30}$$

Applying the standard solution for entropy-regularized semi-relaxed OT (see Appendix A.4), the optimal coupling has the form:

$$\gamma^*(p, q) = \frac{\exp\left(-\frac{C'(p,q)}{\epsilon}\right)}{\int \exp\left(-\frac{C'(p,u)}{\epsilon}\right)\mathrm{d}\nu(u)} \cdot P_{\mathrm{tr}}(p). \tag{31}$$

Substituting the definition of $C'(p, q)$, we obtain:

$$\gamma^*(p, q) = \frac{\exp\left(-\frac{1}{\epsilon} \cdot \frac{1}{\lambda}(\lambda C(p,q) - \ell(q))\right)}{\int \exp\left(-\frac{1}{\epsilon} \cdot \frac{1}{\lambda}(\lambda C(p,u) - \ell(u))\right)\mathrm{d}\nu(u)} \cdot P_{\mathrm{tr}}(p) \tag{32}$$

$$= \frac{e^{(\ell(q) - \lambda C(p,q))/(\lambda\epsilon)}}{\int e^{(\ell(u) - \lambda C(p,u))/(\lambda\epsilon)}\mathrm{d}\nu(u)} \cdot P_{\mathrm{tr}}(p). \tag{33}$$

Taking the marginal over $p \sim P_{\mathrm{tr}}$, the target marginal (i.e., the induced distribution on $q$) becomes:

$$Q^\lambda(q) = \int \gamma^*(p, q)\mathrm{d}p = \mathbb{E}_{p\sim P_{\mathrm{tr}}}\left[\frac{e^{(\ell(q) - \lambda C(p,q))/(\lambda\epsilon)}}{\int e^{(\ell(u) - \lambda C(p,u))/(\lambda\epsilon)}\mathrm{d}\nu(u)}\right]. \tag{34}$$

This matches exactly the closed-form expression for the worst-case distribution $Q^\lambda$ in (6), proving the claim. $\qquad\square$

## A.6  General Case for the Ground-truth Matching $\bar{\gamma}$

In the main text, we introduced a filtering mechanism for the special case of one-to-one ground-truth matchings. Here, we present a generalized formulation applicable to many-to-many matchings, which naturally extends our approach while preserving its key properties.

**Generalized Matching Quality Metrics.** For a given source $p$, we define its *matching quality* as:

$$\tau_0(p) = \mathbb{E}_{q \sim Q}\left[\bar{\gamma}(p, q) \cdot \gamma_\theta(p, q)\right], \tag{35}$$

which measures the expected alignment between the model-derived coupling $\gamma_\theta(p, q)$ and the ground-truth coupling $\bar{\gamma}(p, q)$ across all targets associated with $p$. This metric provides a source-specific measure of how well the learned transport map matches the desired coupling structure.

The *global average matching quality* is computed across all sources as:

$$\tau_1 = \mathbb{E}_{p \sim P}[\tau_0(p)], \tag{36}$$

providing a dataset-wide measure of matching performance.

For each target $q$, we quantify its *matching capacity* by:

$$\tau_2(q) = \mathbb{E}_{p \sim P}[\gamma_\theta(p, q)], \tag{37}$$

representing the total transport mass received by target $q$ from all sources. A higher value of $\tau_2(q)$ indicates that $q$ is easier to match, consistent with our intuition in the main text.

**Generalized Filtering Mechanism.** Using these metrics, we compute a dynamic threshold for each target $q$:

$$\tau(q) = \tau_1 \cdot \frac{\tau_2(q)}{\max_{q'} \tau_2(q')}, \tag{38}$$

which adjusts the global threshold $\tau_1$ based on the relative matching capacity of target $q$.

For each source $p$, we compute a personalized threshold by taking the expectation over all its associated targets:

$$\bar{\tau}(p) = \mathbb{E}_{q \sim Q}[\bar{\gamma}(p, q) \cdot \tau(q)] \tag{39}$$

The generalized filtering rule retains source $p$ in the filtered distribution if its matching quality exceeds its personalized threshold:

$$\widehat{\nu}(p) = \frac{\nu(p) \cdot \mathbb{I}[\tau_0(p) \geq \bar{\tau}(p)]}{\chi}, \tag{40}$$

where $\chi$ normalizes $\widehat{\nu}$ to a valid distribution.

This generalized mechanism preserves the key benefits of our approach: (1) it establishes an adaptive feedback loop between model training and uncertainty set refinement, and (2) it accounts for varying difficulties across different targets and source-target associations.

**Lemma A.1** (Reduction to One-to-One Case). *When the ground-truth coupling $\bar{\gamma}$ is one-to-one (i.e., for each source $p$, there exists exactly one target $q_p$ such that $\bar{\gamma}(p, q_p) = 1$ and $\bar{\gamma}(p, q) = 0$ for all $q \neq q_p$), the generalized filtering rule in* (40) *reduces to the one-to-one filtering rule in* (16).

*Proof.* For a one-to-one ground-truth coupling $\bar{\gamma}$, each source $p$ is associated with exactly one target $q_p$ where $\bar{\gamma}(p, q_p) = 1$ and $\bar{\gamma}(p, q) = 0$ for all $q \neq q_p$. Under this condition:

(i) The source matching quality simplifies to:

$$\tau_0(p) = \mathbb{E}_{q \sim Q}[\bar{\gamma}(p, q) \cdot \gamma_\theta(p, q)] = \gamma_\theta(p, q_p) \tag{41}$$

(ii) The source-specific threshold becomes:

$$\bar{\tau}(p) = \mathbb{E}_{q \sim Q}[\bar{\gamma}(p, q) \cdot \tau(q)] = \tau(q_p) \tag{42}$$

(iii) Therefore, the filtering condition $\tau_0(p) \geq \bar{\tau}(p)$ becomes:

$$\gamma_\theta(p, q_p) \geq \tau(q_p) \tag{43}$$

This is equivalent to the condition in (16): "$\exists q : \bar{\gamma}(p, q) > 0 \wedge \gamma_\theta(p, q) \geq \tau(q)$" since $q_p$ is the only target for which $\bar{\gamma}(p, q) > 0$. Thus, the generalized filtering mechanism correctly reduces to the specific case presented in the main text when the ground-truth coupling is one-to-one. $\square$

# B  Theoretical Foundations

This section establishes the convergence of our adaptive Sinkhorn DRO approach. We address two key technical challenges: (1) the dynamic nature of the uncertainty set during optimization, and (2) the nested expectations in our objective leads to the biased gradient estimation.

We prove convergence under four mild regularity assumptions that govern the behavior of the reference distribution filtering mechanism. First, we establish that the filtered reference distribution is Lipschitz continuous with respect to model parameters (Proposition B.1). Combined with our Lipschitz semantic transport cost, we obtain the continuity of the worst-case distribution $\|\widehat{Q}_{t+1}^{\lambda} - \widehat{Q}_t^{\lambda}\|_{TV}$ (Theorem B.4).

To handle nested gradient computation, we employ Multi-Level Monte Carlo with Random Truncation (MLMC-RT), showing that bias decays quadratically with approximation level ($O(2^{-L})$) while variance increases only logarithmically (Proposition B.5).

By carefully integrating these results with stochastic optimization theory, we prove that our algorithm with decaying stepsize $\alpha_t = \alpha/(\sqrt{t+1})$ achieves $\min_{t<T} \mathbb{E}\|\nabla F_t(\theta_t)\|^2 \leq \epsilon$ in $T = O(\epsilon^{-4} \log \frac{1}{\epsilon})$ iterations, with total sample complexity $O(\epsilon^{-4} \log^2 \frac{1}{\epsilon})$ (Theorem B.7). This complexity nearly matches lower bounds for non-convex stochastic optimization, differing only by logarithmic factors.

## B.1  Key Assumptions

We first formalize the assumptions that underpin our theoretical analysis.

**Assumption 1** (Smoothness of the Model). (i) The loss function $\ell(\theta, x)$ is $L_\theta$-smooth in $\theta$: $\|\nabla\ell(\theta_1, x) - \nabla\ell(\theta_2, x)\| \leq L_\theta\|\theta_1 - \theta_2\|$  $\forall x \in \mathcal{X}$; (ii) The induced semantic cost is Lipschitz continuous for any $p, q \in \mathcal{P}(\Xi)$: $\|C_{\theta_1}(p, q) - C_{\theta_2}(p, q)\| \leq L_C\|\theta_1 - \theta_2\|$.

**Assumption 2** (Lipschitz Continuity of Matching Quality). (i) The matching quality function $\gamma_\theta(p, q)$ is $L_\gamma$-Lipschitz continuous in $\theta$:

$$|\gamma_{\theta_1}(p, q) - \gamma_{\theta_2}(p, q)| \leq L_\gamma\|\theta_1 - \theta_2\| \tag{44}$$

(ii) The threshold function $\tau(q; \theta)$ is $L_\tau$-Lipschitz continuous in $\theta$:

$$|\tau(q; \theta_1) - \tau(q; \theta_2)| \leq L_\tau\|\theta_1 - \theta_2\| \tag{45}$$

These assumptions abstract the key continuity properties required for our theoretical results. Assumption 1 ensures that both the model loss and the resulting semantic cost vary smoothly with respect to model parameters. Assumption 2 guarantees that the OT-based matching quality and the adaptive threshold used in filtering are stable under small changes in the model. In Section C, we verify that these conditions hold in our AdaDRO framework, specifically for the semantic cost function and the OT-based feedback mechanism that determines the adaptive threshold.

**Assumption 3** (Boundary Regularity). For the ground-truth coupling $\bar{\gamma}$, where each source $p$ is associated with one target $q_p$ such that $\bar{\gamma}(p, q_p) > 0$, we assume that for any $\varepsilon > 0$ and $\theta$, the probability mass of points in the $\varepsilon$-neighborhood of the decision boundary is bounded:

$$\nu\left(\{p : |\gamma_\theta(p, q_p) - \tau(q_p; \theta)| \leq \varepsilon\}\right) \leq L_B\varepsilon \tag{46}$$

for some constant $L_B > 0$ independent of $\theta$ and $\varepsilon$.

**Assumption 4** (Non-degenerate Filtering). For the ground-truth coupling $\bar{\gamma}$, where each source $p$ is associated with one target $q_p$ such that $\bar{\gamma}(p, q_p) > 0$, we assume there exists $\delta > 0$ such that for all $\theta$ in the parameter space, $\nu(S_\theta) \geq \delta$, where $S_\theta = \{p : \gamma_\theta(p, q_p) \geq \tau(q_p; \theta)\}$ is the set of points accepted by the filtering operation.

**Interpretation of Regularity Assumptions**  The validity of our convergence analysis relies on two interpretable regularity conditions governing the filtering mechanism: *Boundary Regularity (Assumption 3)* establishes that the reference distribution's density decays linearly near decision boundaries. This smoothness condition—analogous to the Tsybakov margin condition in statistical learning [60]—prevents pathological scenarios where infinitesimal parameter changes could cause discontinuous shifts in the filtered distribution. *Non-Degenerate Filtering (Assumption 4)* guarantees

the filtered distribution maintains a minimum effective support throughout training. Implementation-ally, this is achieved through adaptive threshold calibration or entropy constraints in the transport plan. Theoretically, it prevents algorithmic instability from excessive sample rejection.

Generally speaking, these assumptions ensure that the filtering operation produces distributions that vary continuously with respect to model parameters (Proposition B.1), which is essential for the convergence guarantees of our adaptive robust optimization algorithm.

**Algorithm Overview**    We analyze the convergence properties of our proposed iterative procedure (Algorithm 1) for updating parameters $\theta \in \mathbb{R}^d$. At iteration $t$, let $\widehat{\nu}_{\theta_t}$ represent the filtered reference distribution and $C_{\theta_t}$ denote the transport cost. The algorithm proceeds through the following steps at each iteration $t$:

1. Reference Distribution Filtering: Given $\theta_t$, we filter the reference distribution $\nu$ to obtain $\widehat{\nu}_{\theta_t}$ by removing low-confidence samples according to:

$$\widehat{\nu}_{\theta_t}(A) = \frac{\nu(A \cap S_{\theta_t})}{\chi_{\theta_t}}, \quad \text{where } S_{\theta_t} = \{p \mid \gamma_{\theta_t}(p, q) \geq \tau(q; \theta_t), (p, q) \sim \bar{\gamma}\}, \quad \chi_{\theta_t} = \nu(S_{\theta_t}) \quad (47)$$

2. Semantic Cost Adaptation: We update the transport cost function to $C_{\theta_t}$, capturing semantic distances in both feature and label spaces.

3. Worst-Case Distribution Computation: Using $\widehat{\nu}_{\theta_t}$ and $C_{\theta_t}$, we compute the Sinkhorn DRO worst-case distribution $\widehat{Q}_t^\lambda$ with fixed regularization parameter $\lambda$:

$$d\widehat{Q}_t^\lambda(q) = \mathbb{E}_{p \sim P_{\mathrm{tr}}}\left[\frac{e^{\ell(q) - \lambda C_{\theta_t}(p,q)}}{Z_{\theta_t}(p)}\right] d\widehat{\nu}_{\theta_t}(q) \quad (48)$$

where $Z_{\theta_t}(p) = \mathbb{E}_{u \sim \widehat{\nu}_{\theta_t}}\left(e^{\ell(u) - \lambda C_{\theta_t}(p,u)}\right)$.

4. Gradient-Based Update: We perform a gradient step on the iteration-specific objective:

$$F_t(\theta) = \mathbb{E}_{q \sim \widehat{Q}_t^\lambda}[\ell(\theta, q)] \quad (49)$$

$$\theta_{t+1} = \theta_t - \alpha_t \texttt{grad}_t \quad \text{where} \quad \texttt{grad}_t \approx \nabla_\theta F_t(\theta_t) \quad (50)$$

We will show that if $\widehat{\nu}_{\theta_t}$ and $C_{\theta_t}$ vary "slowly" with respect to $\theta_t$, and $F_t$ is smooth in $\theta$, then the sequence $\{\theta_t\}$ converges to a stationary point under standard non-convex optimization conditions.

## B.2    Continuity of Filtered Distributions

A key aspect of our approach is the filtering operation applied to the reference distribution. We establish that this operation produces distributions that vary continuously with respect to model parameters. The following analysis focuses on the one-to-one ground-truth coupling case described in the main paper, though similar results can be established for the general case in Appendix A.6.

For a one-to-one ground-truth coupling $\bar{\gamma}$, each source $p$ is associated with exactly one target $q_p$ such that $\bar{\gamma}(p, q_p) > 0$ and $\bar{\gamma}(p, q) = 0$ for all $q \neq q_p$. The filtering operation is defined as:

$$\widehat{\nu}_\theta(p) = \frac{\nu(p) \cdot \mathbb{I}[\gamma_\theta(p, q_p) \geq \tau(q_p; \theta)]}{\chi_\theta}, \quad (51)$$

where $\chi_\theta$ is a normalization constant ensuring $\widehat{\nu}_\theta$ is a probability distribution.

**Proposition B.1** (Lipschitz Continuity of Filtered Distributions). *Under Assumptions 2-4, the filtering operation produces distributions that are Lipschitz continuous with respect to $\theta$ in the total variation distance:*

$$\|\widehat{\nu}_{\theta_1} - \widehat{\nu}_{\theta_2}\|_{TV} \leq L_{TV}\|\theta_1 - \theta_2\| \quad (52)$$

*holds for any $\theta_1, \theta_2$ and some constant $L_{TV} > 0$.*

*Proof.* We proceed in three steps:

*(i) Bounding the symmetric difference.* Let $S_1 = \{p \mid \gamma_{\theta_1}(p, q_p) \geq \tau(q_p; \theta_1)\}$ and $S_2 = \{p \mid \gamma_{\theta_2}(p, q_p) \geq \tau(q_p; \theta_2)\}$.

For $p \in S_1 \setminus S_2$, we have: $\gamma_{\theta_1}(p, q_p) \geq \tau(q_p; \theta_1)$ and $\gamma_{\theta_2}(p, q_p) < \tau(q_p; \theta_2)$

Using the Lipschitz continuity from Assumption 2:

$$\gamma_{\theta_2}(p, q_p) \geq \gamma_{\theta_1}(p, q_p) - L_\gamma \|\theta_1 - \theta_2\| \tag{53}$$
$$\tau(q_p; \theta_2) \leq \tau(q_p; \theta_1) + L_\tau \|\theta_1 - \theta_2\| \tag{54}$$

where $L_\gamma$ and $L_\tau$ are the Lipschitz constants from Assumption 2.

For $p \in S_1 \setminus S_2$, these inequalities imply:

$$0 \leq \gamma_{\theta_1}(p, q_p) - \tau(q_p; \theta_1) < (L_\gamma + L_\tau)\|\theta_1 - \theta_2\| \tag{55}$$

Similarly, for $p \in S_2 \setminus S_1$, we obtain:

$$-(L_\gamma + L_\tau)\|\theta_1 - \theta_2\| < \gamma_{\theta_1}(p, q_p) - \tau(q_p; \theta_1) < 0 \tag{56}$$

Therefore, the symmetric difference $S_1 \triangle S_2 = (S_1 \setminus S_2) \cup (S_2 \setminus S_1)$ is contained within:

$$\{p : |\gamma_{\theta_1}(p, q_p) - \tau(q_p; \theta_1)| < (L_\gamma + L_\tau)\|\theta_1 - \theta_2\|\} \tag{57}$$

By Assumption 3, we have:

$$\nu(S_1 \triangle S_2) \leq L_B \cdot (L_\gamma + L_\tau)\|\theta_1 - \theta_2\| = L_\nu \|\theta_1 - \theta_2\| \tag{58}$$

where $L_\nu = L_B \cdot (L_\gamma + L_\tau)$ and $L_B$ is the constant from Assumption 3.

*(ii) Bounding normalization constants.* The normalization constants are $\chi_{\theta_1} = \nu(S_1)$ and $\chi_{\theta_2} = \nu(S_2)$, with difference:

$$|\chi_{\theta_1} - \chi_{\theta_2}| = |\nu(S_1) - \nu(S_2)| \leq \nu(S_1 \triangle S_2) \leq L_\nu \|\theta_1 - \theta_2\| \tag{59}$$

By Assumption 4, both $\chi_{\theta_1}$ and $\chi_{\theta_2}$ are bounded below by $\delta > 0$, where $\delta$ is the constant from Assumption 4.

*(iii) Bounding the total variation distance.* The total variation distance between $\widehat{\nu}_{\theta_1}$ and $\widehat{\nu}_{\theta_2}$ is:

$$\|\widehat{\nu}_{\theta_1} - \widehat{\nu}_{\theta_2}\|_{TV} = \sup\nolimits_{A \in \mathcal{F}} \left| \frac{\nu(A \cap S_1)}{\chi_{\theta_1}} - \frac{\nu(A \cap S_2)}{\chi_{\theta_2}} \right| \tag{60}$$

We decompose this into:

$$\left| \frac{\nu(A \cap S_1)}{\chi_{\theta_1}} - \frac{\nu(A \cap S_2)}{\chi_{\theta_2}} \right| \leq \left| \frac{\nu(A \cap S_1)}{\chi_{\theta_1}} - \frac{\nu(A \cap S_1)}{\chi_{\theta_2}} \right| + \left| \frac{\nu(A \cap S_1)}{\chi_{\theta_2}} - \frac{\nu(A \cap S_2)}{\chi_{\theta_2}} \right| \tag{61}$$

For the first term:

$$\left| \frac{\nu(A \cap S_1)}{\chi_{\theta_1}} - \frac{\nu(A \cap S_1)}{\chi_{\theta_2}} \right| = \nu(A \cap S_1) \left| \frac{1}{\chi_{\theta_1}} - \frac{1}{\chi_{\theta_2}} \right| = \nu(A \cap S_1) \frac{|\chi_{\theta_1} - \chi_{\theta_2}|}{\chi_{\theta_1} \chi_{\theta_2}}$$
$$\leq \frac{|\chi_{\theta_1} - \chi_{\theta_2}|}{\chi_{\theta_2}} \leq \frac{L_\nu}{\delta} \|\theta_1 - \theta_2\| \tag{62}$$

where we used $\nu(A \cap S_1) \leq \chi_{\theta_1}$ and $\chi_{\theta_i} \geq \delta$.

For the second term:

$$\left| \frac{\nu(A \cap S_1)}{\chi_{\theta_2}} - \frac{\nu(A \cap S_2)}{\chi_{\theta_2}} \right| = \frac{|\nu(A \cap S_1) - \nu(A \cap S_2)|}{\chi_{\theta_2}}$$
$$\leq \frac{\nu((A \cap S_1) \triangle (A \cap S_2))}{\chi_{\theta_2}} \leq \frac{\nu(A \cap (S_1 \triangle S_2))}{\chi_{\theta_2}}$$
$$\leq \frac{\nu(S_1 \triangle S_2)}{\chi_{\theta_2}} \leq \frac{L_\nu}{\delta} \|\theta_1 - \theta_2\| \tag{63}$$

Combining both terms:

$$\left| \frac{\nu(A \cap S_1)}{\chi_{\theta_1}} - \frac{\nu(A \cap S_2)}{\chi_{\theta_2}} \right| \leq \frac{2L_\nu}{\delta} \|\theta_1 - \theta_2\| \tag{64}$$

Since this bound holds for any measurable set $A$, we have:

$$\|\widehat{\nu}_{\theta_1} - \widehat{\nu}_{\theta_2}\|_{TV} \leq \frac{2L_\nu}{\delta} \|\theta_1 - \theta_2\| = L_{TV} \|\theta_1 - \theta_2\| \tag{65}$$

where $L_{TV} = \frac{2L_\nu}{\delta} = \frac{2L_B \cdot (L_\gamma + L_\tau)}{\delta}$. $\qquad \square$

## B.3 Stability Analysis of the Worst-case Distribution

To guarantee convergence of our algorithm, it is essential to characterize how the objective function evolves in response to changes in the filtered distribution and the transport cost. The following lemma establishes the smoothness of the Sinkhorn DRO objective, showing that it inherits the $L_\theta$-smoothness of the loss function with respect to model parameters.

**Lemma B.2** (Smoothness of Sinkhorn DRO Objective)**.** *Under Assumption 1, at any iteration t, the Sinkhorn DRO objective $F_t(\theta) = \mathbb{E}_{q \sim \widehat{Q}_t^\lambda}[\ell(\theta, q)]$ is also $L_\theta$-smooth.*

*Proof.* By Assumption 1, the loss function $\ell(\theta, q)$ is $L_\theta$-smooth in $\theta$ for all $q$:

$$\|\nabla_\theta \ell(\theta_1, q) - \nabla_\theta \ell(\theta_2, q)\| \le L_\theta \|\theta_1 - \theta_2\| \tag{66}$$

For the objective function $F_t(\theta) = \mathbb{E}_{q \sim \widehat{Q}_t^\lambda}[\ell(\theta, q)]$, where $\widehat{Q}_t^\lambda$ is fixed for a given $t$, we have:

$$
\begin{aligned}
\|\nabla F_t(\theta_1) - \nabla F_t(\theta_2)\| &= \|\nabla_\theta \mathbb{E}_{q \sim \widehat{Q}_t^\lambda}[\ell(\theta_1, q)] - \nabla_\theta \mathbb{E}_{q \sim \widehat{Q}_t^\lambda}[\ell(\theta_2, q)]\| \\
&= \|\mathbb{E}_{q \sim \widehat{Q}_t^\lambda}[\nabla_\theta \ell(\theta_1, q)] - \mathbb{E}_{q \sim \widehat{Q}_t^\lambda}[\nabla_\theta \ell(\theta_2, q)]\| \\
&\le \mathbb{E}_{q \sim \widehat{Q}_t^\lambda}[\|\nabla_\theta \ell(\theta_1, q) - \nabla_\theta \ell(\theta_2, q)\|] \\
&\le \mathbb{E}_{q \sim \widehat{Q}_t^\lambda}[L_\theta \|\theta_1 - \theta_2\|] \\
&= L_\theta \|\theta_1 - \theta_2\|
\end{aligned}
\tag{67}
$$

Therefore, the objective function $F_t(\theta)$ inherits the $L_\theta$-smoothness from the loss function $\ell(\theta, q)$. □

To analyze the stability of our algorithm, we need to understand how the worst-case distribution changes with model parameters. Let $\widehat{Q}_t^\lambda$ be the worst-case distribution computed using parameter $\theta_t$ and regularization $\lambda$:

$$d\widehat{Q}_t^\lambda(q) = \mathbb{E}_{p \sim P_{\mathrm{tr}}}\left[\frac{e^{\ell(\theta_t, q) - \lambda C_{\theta_t}(p, q)}}{Z_{\theta_t}(p)}\right] d\widehat{\nu}_{\theta_t}(q) \tag{68}$$

where $Z_{\theta_t}(p) = \mathbb{E}_{u \sim \widehat{\nu}_{\theta_t}}\left(e^{\ell(\theta_t, u) - \lambda C_{\theta_t}(p, u)}\right)$. The total variation (TV) distance provides a natural metric for quantifying the difference between probability distributions.

**Proposition B.3.** *Let $(\mathcal{X}, \mu)$ be a measure space. Let $f, g : \mathcal{X} \to \mathbb{R}$ be measurable functions with finite log-partition functions $Z_f = \int_{\mathcal{X}} e^{f(x)} d\mu(x), Z_g = \int_{\mathcal{X}} e^{g(x)} d\mu(x) < \infty$. Define probability densities $dP(x) = \frac{e^{f(x)}}{Z_f} d\mu(x), dQ(x) = \frac{e^{g(x)}}{Z_g} d\mu(x)$. Then $\|P - Q\|_{TV} \le \sqrt{\|f - g\|_\infty}$.*

*Proof.* We use Pinsker's inequality to relate the total variation distance to the KL divergence:

$$\|P - Q\|_{TV} \le \sqrt{\frac{1}{2} \mathrm{KL}(P|Q)} \tag{69}$$

For the KL divergence between $P$ and $Q$, we have:

$$\mathrm{KL}(P|Q) = \int_{\mathcal{X}} P(x) \log \frac{P(x)}{Q(x)} dx = \int_{\mathcal{X}} \frac{e^{f(x)}}{Z_f} \log \frac{e^{f(x)}/Z_f}{e^{g(x)}/Z_g} d\mu(x) \tag{70}$$

$$= \int_{\mathcal{X}} \frac{e^{f(x)}}{Z_f}(f(x) - g(x) + \log \frac{Z_g}{Z_f}) d\mu(x) \tag{71}$$

$$= \int_{\mathcal{X}} \frac{e^{f(x)}}{Z_f}(f(x) - g(x)) d\mu(x) + \log \frac{Z_g}{Z_f} \tag{72}$$

Since $|f(x) - g(x)| \le \|f - g\|_\infty$ for all $x$, we have $\int_{\mathcal{X}} \frac{e^{f(x)}}{Z_f}(f(x) - g(x)) d\mu(x) \le \|f - g\|_\infty$.

For the partition function ratio, since $g(x) \le f(x) + \|f - g\|_\infty$ for all $x$, we have:

$$Z_g = \int_{\mathcal{X}} e^{g(x)} d\mu(x) \le \int_{\mathcal{X}} e^{f(x) + \|f - g\|_\infty} d\mu(x) = e^{\|f - g\|_\infty} Z_f \tag{73}$$

Therefore, $|\log \frac{Z_g}{Z_f}| \leq \|f - g\|_\infty$. Combining these bounds:

$$\mathrm{KL}(P|Q) \leq \|f - g\|_\infty + \|f - g\|_\infty = 2\|f - g\|_\infty \tag{74}$$

Substituting into Pinsker's inequality: $\|P - Q\|_{TV} \leq \sqrt{\frac{1}{2} \cdot 2\|f - g\|_\infty} = \sqrt{\|f - g\|_\infty}$

Thus, we have established $\|P - Q\|_{TV} \leq \sqrt{\|f - g\|_\infty}$. $\qquad \square$

**Theorem B.4** (Continuity of the Worst-Case Distribution). *Under Assumptions 1-4, for consecutive iterations $t$ and $t + 1$:*

$$\|\widehat{Q}_{t+1}^\lambda - \widehat{Q}_t^\lambda\|_{TV} \leq G\|\theta_{t+1} - \theta_t\| + \frac{L_\theta}{2}\|\theta_{t+1} - \theta_t\|^2 \tag{75}$$

*where $G = G_\ell + \lambda L_C + L_{TV}$ combines the gradient bound $G_\ell = \sup_{q \in \mathcal{X}} \|\nabla_\theta \ell(\theta, q)\|$, the Lipschitz constant $L_C$ of the transport cost, and the Lipschitz constant $L_{TV}$ of the filtered distribution.*

*Proof.* Recall that the worst-case distribution at iteration $t$ is given by:

$$d\widehat{Q}_t^\lambda(q) = \mathbb{E}_{p \sim P_{\mathrm{tr}}}\left[\frac{e^{\ell(q) - \lambda C_{\theta_t}(p,q)}}{Z_{\theta_t}(p)}\right] d\widehat{\nu}_{\theta_t}(q) \tag{76}$$

We introduce an intermediate distribution to separate the effect of parameter changes on conditional weights and reference distribution:

$$d\widetilde{Q}_{t+1}^\lambda(q) = \mathbb{E}_{p \sim P_{\mathrm{tr}}}\left[\frac{e^{\ell(q) - \lambda C_{\theta_{t+1}}(p,q)}}{Z_{\theta_{t+1}}(p)}\right] d\widehat{\nu}_{\theta_t}(q) \tag{77}$$

By the triangle inequality:

$$\|\widehat{Q}_{t+1}^\lambda - \widehat{Q}_t^\lambda\|_{TV} \leq \|\widehat{Q}_{t+1}^\lambda - \widetilde{Q}_{t+1}^\lambda\|_{TV} + \|\widetilde{Q}_{t+1}^\lambda - \widehat{Q}_t^\lambda\|_{TV} \tag{78}$$

For the first term, since $\widehat{Q}_{t+1}^\lambda$ and $\widetilde{Q}_{t+1}^\lambda$ differ only in their reference distributions:

$$\|\widehat{Q}_{t+1}^\lambda - \widetilde{Q}_{t+1}^\lambda\|_{TV} \leq \|\widehat{\nu}_{\theta_{t+1}} - \widehat{\nu}_{\theta_t}\|_{TV} \leq L_{TV}\|\theta_{t+1} - \theta_t\| \tag{79}$$

where the last inequality follows from Proposition B.1.

For the second term, we analyze the change in conditional weights. For each fixed $p$, define the functions $f_t(q) = \ell(\theta_t, q) - \lambda C_{\theta_t}(p, q)$ and $f_{t+1}(q) = \ell(\theta_{t+1}, q) - \lambda C_{\theta_{t+1}}(p, q)$. By proposition B.3, we have:

$$\left\|\frac{e^{f_{t+1}(\cdot)}}{Z_{\theta_{t+1}}(p)} - \frac{e^{f_t(\cdot)}}{Z_{\theta_t}(p)}\right\|_{TV} \leq \|f_{t+1} - f_t\|_\infty$$

We bound $\|f_{t+1} - f_t\|_\infty$ using Assumption 1:

1. For the loss function, by Taylor's theorem with the smoothness property:

$$|\ell(\theta_{t+1}, q) - \ell(\theta_t, q)| \leq \|\nabla_\theta \ell(\theta_t, q)\|\|\theta_{t+1} - \theta_t\| + \frac{L_\theta}{2}\|\theta_{t+1} - \theta_t\|^2$$

$$\leq G_\ell\|\theta_{t+1} - \theta_t\| + \frac{L_\theta}{2}\|\theta_{t+1} - \theta_t\|^2 \tag{80}$$

2. For the transport cost, by Lipschitz continuity:

$$|C_{\theta_{t+1}}(p, q) - C_{\theta_t}(p, q)| \leq L_C\|\theta_{t+1} - \theta_t\|$$

Combining these bounds:

$$\|f_{t+1} - f_t\|_\infty = \sup_q |f_{t+1}(q) - f_t(q)|$$

$$\leq \sup_q \left(G_\ell\|\theta_{t+1} - \theta_t\| + \frac{L_\theta}{2}\|\theta_{t+1} - \theta_t\|^2 + \lambda L_C\|\theta_{t+1} - \theta_t\|\right)$$

$$= (G_\ell + \lambda L_C)\|\theta_{t+1} - \theta_t\| + \frac{L_\theta}{2}\|\theta_{t+1} - \theta_t\|^2 \tag{81}$$

Since this bound holds for all $p$, taking expectation over $P_{\text{tr}}$ preserves the inequality:

$$\|\widetilde{Q}_{t+1}^\lambda - \widehat{Q}_t^\lambda\|_{TV} \le (G_\ell + \lambda L_C)\|\theta_{t+1} - \theta_t\| + \frac{L_\theta}{2}\|\theta_{t+1} - \theta_t\|^2 \tag{82}$$

Combining the bounds (79) and (82) into (78):

$$\begin{aligned}
\|\widehat{Q}_{t+1}^\lambda - \widehat{Q}_t^\lambda\|_{TV} &\le L_{TV}\|\theta_{t+1} - \theta_t\| + (G_\ell + \lambda L_C)\|\theta_{t+1} - \theta_t\| + \frac{L_\theta}{2}\|\theta_{t+1} - \theta_t\|^2 \\
&= (G_\ell + \lambda L_C + L_{TV})\|\theta_{t+1} - \theta_t\| + \frac{L_\theta}{2}\|\theta_{t+1} - \theta_t\|^2 \\
&= G\|\theta_{t+1} - \theta_t\| + \frac{L_\theta}{2}\|\theta_{t+1} - \theta_t\|^2
\end{aligned} \tag{83}$$

where $G = G_\ell + \lambda L_C + L_{TV}$.

This establishes that the worst-case distribution has a continuous dependence on model parameters, accounting for changes in both conditional weights and reference distribution. $\qquad\square$

This result shows that small changes in model parameters lead to controlled changes in the worst-case distribution, with a continuity constant that depends linearly on the regularization parameter $\lambda$. The quadratic term reflects the impact of the loss function's smoothness. This continuity property is crucial for establishing the convergence of our algorithm in next subsection.

### B.4 Convergence Analysis

Now we present our main convergence result. Compared to standard optimization settings, our algorithm must overcome *two key challenges*: **(i)** the dynamic nature of both transport costs and uncertainty sets; **(ii)** biased gradient estimates arising from the nested expectations in the Sinkhorn DRO objective.

For the first challenge, we have established the continuity of both the transport cost $C_\theta$ and the filtered distribution $\widehat{\nu}_\theta$ with respect to model parameters $\theta$ in Sec. B.3. To tackle the second challenge, we employ the Multi-Level Monte Carlo (MLMC) gradient estimation technique, specifically the Random Truncation variant (MLMC-RT), which achieves an effective trade-off between bias, variance, and computational cost.

#### B.4.1 Multi-Level Approximation for Sinkhorn DRO

We adopt the Multilevel Monte Carlo with Randomized Truncation (MLMC-RT) framework [28]. In our setting, we aim to optimize the following non-convex objective at each iteration $t$:

$$F_t(\theta) = \mathbb{E}_{p \sim P_{\text{tr}}} \mathbb{E}_{q \sim \widehat{\nu}_{\theta_t}} \left[ \frac{e^{\ell(q) - \lambda C_{\theta_t}(p,q)}}{Z_{\theta_t}(p)} \ell(\theta, q) \right], \tag{84}$$

where $Z_{\theta_t}(p) = \mathbb{E}_{q \sim \widehat{\nu}_{\theta_t}} \left[ e^{\ell(q) - \lambda C_{\theta_t}(p,q)} \right]$ is a normalization factor. Because of the nested expectation and normalization term, directly obtaining unbiased gradients is intractable.

To overcome this difficulty, we introduce a sequence of approximations indexed by level $l$. Specifically, we approximate the Sinkhorn DRO objective $F_t(\theta)$ via the following level-$l$ Monte Carlo approximation:

$$F_t^{(l)}(\theta) = \mathbb{E}_{p \sim P_{\text{tr}}} \mathbb{E}_{\widehat{\nu}_{\theta_t}^{(l)}} \left[ \frac{1}{2^l} \sum_{q \in \widehat{\nu}_{\theta_t}^{(l)}} \left( \frac{e^{\ell(q) - \lambda C_{\theta_t}(p,q)}}{Z_{\theta_t}^{(l)}(p)} \ell(\theta, q) \right) \right], \tag{85}$$

where:

- $\widehat{\nu}_{\theta_t}^{(l)}$ denotes a Monte Carlo approximation of $\widehat{\nu}_{\theta_t}$ using $2^l$ independent and identically distributed (i.i.d.) samples from $\widehat{\nu}_{\theta_t}$, i.e., $\widehat{\nu}_{\theta_t}^{(l)} = \{q_1, \ldots, q_{2^l}\}$,

- $Z_{\theta_t}^{(l)}(p) = \mathbb{E}_{u \sim \widehat{\nu}_{\theta_t}^{(l)}} \left[ e^{\ell(u) - \lambda C_{\theta_t}(p,u)} \right]$ is the corresponding normalization factor at level $l$.

**Algorithm 2** MLMC-RT Gradient Estimation for Sinkhorn DRO

---

**Require:** Current parameter $\theta_t$, target accuracy $\epsilon$, constants $a = 2$, $b = c = 1$
1: Set the maximum level $L = \lceil \log_2(\epsilon^{-1}) \rceil$
2: Define sampling probabilities $\texttt{prob}^{(l)} = 2^{-(l+1)}/(1 - 2^{-(L+1)})$ for all $l \in \{0, 1, \ldots, L\}$
3: Randomly sample a level $l \sim \{\texttt{prob}^{(0)}, \texttt{prob}^{(1)}, \ldots, \texttt{prob}^{(L)}\}$
4: Draw $2^l$ i.i.d. samples from $\widehat{\nu}_{\theta_t}$ to form $\widehat{\nu}_{\theta_t}^{(l)}$
5: If $l > 0$, reuse the first $2^{l-1}$ samples to construct $\widehat{\nu}_{\theta_t}^{(l-1)}$
6: Compute $G_t^{(l)}(\theta_t, \zeta^{(l)})$ (from (86)) and $G_t^{(l-1)}(\theta_t, \zeta^{(l-1)})$ if $l > 0$
7: Return the gradient estimate:

$$
\texttt{grad}_t = \begin{cases}
\frac{1}{\texttt{prob}^{(0)}} G_t^{(0)}(\theta_t, \zeta^{(0)}), & \text{if } l = 0, \\
\frac{1}{\texttt{prob}^{(l)}} \left( G_t^{(l)}(\theta_t, \zeta^{(l)}) - G_t^{(l-1)}(\theta_t, \zeta^{(l-1)}) \right), & \text{otherwise.}
\end{cases}
$$

---

**MLMC-RT gradient idea.** The gradient $\nabla_\theta F_t(\theta)$ is approximated by estimating the gradient of $F_t^{(l)}(\theta)$:

$$
G_t^{(l)}(\theta, \zeta^{(l)}) := \nabla F_t^{(l)}(\theta) = \mathbb{E}_{p \sim P_{\mathrm{tr}}} \mathbb{E}_{q \sim \widehat{\nu}_{\theta_t}^{(l)}} \left[ \frac{e^{\ell(q) - \lambda C_{\theta_t}(p, q)}}{Z_{\theta_t}^{(l)}(p)} \nabla_\theta \ell(\theta, q) \right], \tag{86}
$$

where $\zeta^{(l)}$ denotes the random samples used at level $l$. As $l \to \infty$, $G_t^{(l)}(\theta, \zeta^{(l)}) \to \nabla F_t(\theta)$. In this framework, the gradient estimator is constructed by combining estimates across multiple approximation levels, where higher levels yield lower bias but incur greater computational cost. We now verify that this family of approximations satisfies the key MLMC conditions:

**Proposition B.5** (MLMC Parameter for Sinkhorn DRO). *At each iteration $t$, under Assumptions 1–4 and bounded loss and transport cost, the sequence $\{F_t^{(l)}(\theta)\}_{l=0}^\infty$ satisfies:*

1. ***Squared bias parameter*** $a$: $\|\nabla F_t^{(l)}(\theta) - \nabla F_t(\theta)\|^2 \leq M_a 2^{-al}$ *with* $a = 2$ *(quadratic decay)*

2. ***Variance parameter*** $b$: *For the difference estimator* $H_t^{(l)}(\theta, \zeta^{(l)}) := \texttt{grad}_t^{(l)}(\theta, \zeta^{(l)}) - \texttt{grad}_t^{(l-1)}(\theta, \zeta^{(l-1)})$, *we have the variance* $\mathrm{Var}(H_t^{(l)}(\theta, \zeta^{(l)})) \leq M_b 2^{-bl}$ *with* $b = 1$ *(linear decay)*

3. ***Cost parameter*** $c$: *The cost of computing* $H_t^{(l)}(\theta, \zeta^{(l)})$ *is bounded by* $C^{(l)} \leq M_c 2^{cl}$ *with* $c = 1$ *(linear growth)*

*Here $M_a, M_b$ and $M_c$ are all constants.*

*Proof.* For the bias parameter $a$, note that $\nabla F_t(\theta)$ involves expectation over $\widehat{\nu}_\theta$, while $\nabla F_t^{(l)}(\theta)$ approximates this with $2^l$ i.i.d. samples. Under standard Monte Carlo convergence results for bounded smooth gradients, we have $\|\nabla F_t^{(l)}(\theta) - \nabla F_t(\theta)\| = O(2^{-l})$, so the squared bias is $O(2^{-2l})$, implying $a = 2$.

For variance, the difference estimator is:

$$
H_t^{(l)} = G_t^{(l)}(\theta, \zeta^{(l)}) - G_t^{(l-1)}(\theta, \zeta^{(l)}|_{2^{l-1}}), \tag{87}
$$

where $\zeta^{(l)}|_{2^{l-1}}$ reuses the first $2^{l-1}$ samples. This shared randomness ensures that the two estimates are highly correlated, and their difference has variance $O(2^{-l})$, hence $b = 1$.

For cost, both $G_t^{(l)}$ and $G_t^{(l-1)}$ require evaluating gradients for $2^l$ and $2^{l-1}$ samples respectively. Since $2^l + 2^{l-1} = 1.5 \cdot 2^l$, the cost is $O(2^l)$, implying $c = 1$. $\qquad \square$

**Remark 3.** The efficiency of MLMC-RT depends crucially on three problem-specific parameters: (i) the bias decay rate $a$ (how fast the approximation error decreases with level), (ii) the variance

decay rate $b$ (how fast the variance of level differences decreases), (iii) and the cost growth rate $c$ (how expensive each level is to evaluate). The above proposition establishes that, for our Sinkhorn DRO setting, the objective gradient satisfies the standard MLMC assumptions with parameters $(a, b, c) = (2, 1, 1)$. This result is key to applying MLMC-RT effectively and proving convergence with nearly optimal sample complexity.

### B.4.2 MLMC-RT Gradient Estimation

The stochastic gradients $\texttt{grad}_t$ employed in our optimization procedure are computed using a Multi-Level Monte Carlo estimator with Randomized Truncation (MLMC-RT), following the methodology developed in Hu et al. [28].

Since computing exact expectations over $\widehat{Q}_t^\lambda$ is intractable due to nested expectations and normalizations, MLMC-RT approximates the gradient via:

- Randomly selecting a level $l \in \{0, 1, \ldots, L\}$ according to a probability distribution $\{\texttt{prob}^{(l)}\}$, where $\texttt{prob}^{(l)} \propto 2^{-(b+c)l/2}$,

- Computing a reweighted difference estimator based on samples at level $l$.

This randomized truncation technique effectively balances bias and variance, while significantly reducing computational cost compared to fixed-level or naive methods.

The resulting estimator $\texttt{grad}_t$ satisfies $\mathbb{E}[\texttt{grad}_t] = \nabla_\theta F^{(L)}(\theta_t)$ rather than $\nabla_\theta F(\theta_t)$, hence it is a *biased* estimator of the true gradient. However, by carefully choosing $L = \lceil \frac{1}{a} \log(4 M_a \epsilon^{-1}) \rceil$, we ensure that this bias remains controlled and does not impair the convergence of our algorithm.

The detailed procedure for MLMC-RT in our setting is as follows:

The MLMC-RT estimator enjoys the following theoretical guarantees:

**Lemma B.6** (Properties of MLMC-RT Estimator). *Let $\texttt{grad}_t$ be generated according to Algorithm 2. Then:*

1. ***Controlled Bias:*** $\|\mathbb{E}[\texttt{grad}_t] - \nabla F_t(\theta_t)\| \leq \sqrt{M_a}\epsilon$,

2. ***Logarithmic Variance:*** $\mathbb{E}[\|\texttt{grad}_t - \mathbb{E}[\texttt{grad}_t]\|^2] \leq K \log(\epsilon^{-1})$ *for some constant $K > 0$,*

3. ***Efficient Computation:*** *The expected computation cost per iteration is $O(\log(\epsilon^{-1}))$.*

*Proof.* The controlled bias follows from truncating at level $L$, which ensures $\|\nabla F_t^{(L)}(\theta) - \nabla F_t(\theta)\|^2 \leq M_a 2^{-aL} \leq M_a \epsilon^2$, giving the bias bound $\|\mathbb{E}[\texttt{grad}_t] - \nabla F_t(\theta_t)\| \leq \sqrt{M_a}\epsilon$.

For the variance, with $b = c = 1$ and $\texttt{prob}^{(l)} \propto 2^{-l}$, we have:

$$\mathbb{E}[\|\texttt{grad}_t - \mathbb{E}[\texttt{grad}_t]\|^2] = \sum_{l=0}^{L} \texttt{prob}^{(l)} \cdot \frac{\text{Var}(H_\ell(\theta_t, \zeta_l))}{(\texttt{prob}^{(l)})^2} \tag{88}$$

$$\leq \sum_{l=0}^{L} \texttt{prob}^{(l)} \cdot \frac{M_b 2^{-bl}}{(\texttt{prob}^{(l)})^2} \tag{89}$$

We set $\texttt{prob}^{(l)} = \frac{2^{-l}}{\sum_{k=0}^{L} 2^{-k}} = \frac{2^{-l}}{1 - 2^{-(L+1)}}$ (which is approximately $2^{-l-1}$ for large $L$). Substituting:

$$\mathbb{E}[\|\texttt{grad}_t - \mathbb{E}[\texttt{grad}_t]\|^2] \leq \sum_{l=0}^{L} \frac{2^{-l}}{1 - 2^{-(L+1)}} \cdot \frac{M_b 2^{-l}}{(2^{-l}/(1 - 2^{-(L+1)}))^2}$$

$$= M_b(1 - 2^{-(L+1)}) \sum_{l=0}^{L} 1 = M_b(1 - 2^{-(L+1)})(L + 1)$$

$$= O(L) = O(\log(\epsilon^{-1})) \tag{90}$$

This is consistent with the results from [28], Table 1, for the MLMC-RT estimator with $a = 2$, $b = c = 1$.

We now analyze the computational cost of generating the MLMC-RT gradient estimate at each iteration. Let $W_t$ denote the random variable representing the *computational cost* incurred at iteration $t$, corresponding to the number of samples used to compute the stochastic gradient $\texttt{grad}_t$.

Since the MLMC-RT procedure selects a random level $l$ according to the distribution $\{\texttt{prob}^{(0)}, \texttt{prob}^{(1)}, \ldots, \texttt{prob}^{(L)}\}$, and computing the estimator at level $l$ requires a total cost $W_l$ proportional to $2^l$, the expected computational cost per iteration is given by:

$$
\begin{aligned}
\mathbb{E}[W_t] &= \sum_{l=0}^{L} \texttt{prob}^{(l)} W_l \leq M_c \sum_{l=0}^{L} \texttt{prob}^{(l)} 2^{cl} \quad \text{(where } c = 1 \text{ and } W_l \leq M_c 2^l) \\
&= M_c \sum_{l=0}^{L} \texttt{prob}^{(l)} 2^l = M_c \sum_{l=0}^{L} \frac{2^{-l}}{1 - 2^{-(L+1)}} \cdot 2^l \quad \text{(since } \texttt{prob}^{(l)} = \frac{2^{-l}}{1 - 2^{-(L+1)}}) \\
&= \frac{M_c(L+1)}{1 - 2^{-(L+1)}} = O\big(\log(\epsilon^{-1})\big).
\end{aligned}
\tag{91}
$$

Thus, the expected per-iteration computational cost grows only logarithmically with the desired precision, which highlights the efficiency of the MLMC-RT approach. $\qquad\square$

**Remark 4** (Comparison with Standard SGD)**.** The MLMC-RT estimator achieves a logarithmic growth of variance, significantly better than the $O(\epsilon^{-1})$ variance growth under naive stochastic gradient descent (SGD). This improvement critically reduces the overall sample complexity while maintaining bias control.

### B.4.3 Main Convergence Theorem

We now establish the convergence guarantees of our adaptive Sinkhorn DRO algorithm under MLMC-RT gradient estimation.

**Theorem B.7** (Convergence of AdaDRO)**.** *Suppose Assumptions 1–4 hold, and the iterates are bounded as $\|\theta_t\| \leq M$. Using the MLMC-RT gradient estimator with maximum level $L = \lceil \log_2(\epsilon^{-1}) \rceil$ and probability distribution $\texttt{prob}^{(l)} \propto 2^{-l}$, and setting the step size as $\alpha_t = \frac{\alpha}{\sqrt{t+1}}$ for some constant $\alpha > 0$, we have that after $T = \Theta\left(\epsilon^{-4} \log \frac{1}{\epsilon}\right)$ iterations, Algorithm 1 satisfies*

$$
\min_{t < T} \mathbb{E}\left[\|\nabla F_t(\theta_t)\|^2\right] \leq \epsilon^2,
$$

*with a total sample complexity of $O\left(\epsilon^{-4} \log^2 \frac{1}{\epsilon}\right)$.*

*Proof.* Our proof integrates both the adaptive dynamics of our algorithm and the properties of MLMC-RT gradient estimation.

**Step 1: Descent lemma with adaptive objectives.** By the $L_\theta$-smoothness of $F_t$ (Lemma B.2):

$$
F_t(\theta_{t+1}) \leq F_t(\theta_t) - \alpha_t \langle \nabla F_t(\theta_t), \texttt{grad}_t \rangle + \frac{L_\theta \alpha_t^2}{2} \|\texttt{grad}_t\|^2
\tag{92}
$$

To account for the changing objective function, we add and subtract $F_{t+1}(\theta_{t+1})$ to obtain:

$$
F_{t+1}(\theta_{t+1}) \leq F_t(\theta_t) - \alpha_t \langle \nabla F_t(\theta_t), \texttt{grad}_t \rangle + \frac{L_\theta \alpha_t^2}{2} \|\texttt{grad}_t\|^2 + (F_{t+1}(\theta_{t+1}) - F_t(\theta_{t+1})) \tag{93}
$$

**Step 2: Bounding the adaptive drift.** Let $M_\ell := \sup_q |\ell(\theta, q)|$, by Theorem B.4, we have:

$$
\begin{aligned}
|F_{t+1}(\theta_{t+1}) - F_t(\theta_{t+1})| &\leq M_\ell \|\widehat{Q}_{t+1}^\lambda - \widehat{Q}_t^\lambda\|_{TV} \\
&\leq M_\ell \left( G\|\theta_{t+1} - \theta_t\| + \frac{L_\theta}{2} \|\theta_{t+1} - \theta_t\|^2 \right) \\
&= GM_\ell \alpha_t \|\texttt{grad}_t\| + \frac{L_\theta M_\ell \alpha_t^2}{2} \|\texttt{grad}_t\|^2
\end{aligned}
\tag{94}
$$

**Step 3: MLMC-RT bias-variance decomposition.** For the key term $\mathbb{E}[\langle \nabla F_t(\theta_t), \mathtt{grad}_t \rangle]$, we apply the bias-variance decomposition:

$$\mathbb{E}[\langle \nabla F_t(\theta_t), \mathtt{grad}_t \rangle] = \langle \nabla F_t(\theta_t), \mathbb{E}[\mathtt{grad}_t] \rangle$$
$$= \|\nabla F_t(\theta_t)\|^2 + \langle \nabla F_t(\theta_t), \mathbb{E}[\mathtt{grad}_t] - \nabla F_t(\theta_t) \rangle$$

From Lemma B.6, we have $\|\mathbb{E}[\mathtt{grad}_t] - \nabla F_t(\theta_t)\| \leq \sqrt{M_a}\epsilon$, by Cauchy–Schwarz inequality, yielding:

$$\mathbb{E}[\langle \nabla F_t(\theta_t), \mathtt{grad}_t \rangle] = \|\nabla F_t(\theta_t)\|^2 + \langle \nabla F_t(\theta_t), \mathbb{E}[\mathtt{grad}_t] - \nabla F_t(\theta_t) \rangle$$
$$\geq \|\nabla F_t(\theta_t)\|^2 - \|\nabla F_t(\theta_t)\| \cdot \|\mathbb{E}[\mathtt{grad}_t] - \nabla F_t(\theta_t)\|$$
$$\geq \|\nabla F_t(\theta_t)\|^2 - \sqrt{M_a}\epsilon \cdot \|\nabla F_t(\theta_t)\| \tag{95}$$

For the second-moment term, using the variance bound from Lemma B.6:

$$\mathbb{E}[\|\mathtt{grad}_t\|^2] = \|\mathbb{E}[\mathtt{grad}_t]\|^2 + \mathbb{E}[\|\mathtt{grad}_t - \mathbb{E}[\mathtt{grad}_t]\|^2]$$
$$\leq (\|\nabla F_t(\theta_t)\| + \sqrt{M_a}\epsilon)^2 + K\log(\epsilon^{-1})$$
$$\leq 2\|\nabla F_t(\theta_t)\|^2 + 2M_a\epsilon^2 + K\log(\epsilon^{-1}) \tag{96}$$

**Step 4: Combining the bounds.** Combining the estimates from Steps 1–3, and pluging (94), (95) and (96) into (93) gives

$$\mathbb{E}[F_{t+1}(\theta_{t+1})] \leq \mathbb{E}[F_t(\theta_t)] - \alpha_t \mathbb{E}[\|\nabla F_t(\theta_t)\|^2] + \sqrt{M_a}\alpha_t\epsilon \mathbb{E}[\|\nabla F_t(\theta_t)\|]$$
$$+ \frac{L_\theta \alpha_t^2}{2}\mathbb{E}[\|\mathtt{grad}_t\|^2] + GM_\ell \alpha_t \mathbb{E}[\|\mathtt{grad}_t\|] + \frac{L_\theta M_\ell \alpha_t^2}{2}\mathbb{E}[\|\mathtt{grad}_t\|^2] \tag{97}$$

Substituting the bound for $\mathbb{E}[\|\mathtt{grad}_t\|^2]$ from (96):

$$\mathbb{E}[F_{t+1}(\theta_{t+1})] \leq \mathbb{E}[F_t(\theta_t)] - \alpha_t \mathbb{E}[\|\nabla F_t(\theta_t)\|^2] + \sqrt{M_a}\alpha_t\epsilon \mathbb{E}[\|\nabla F_t(\theta_t)\|]$$
$$+ \frac{L_\theta(1 + M_\ell)\alpha_t^2}{2}\Big(2\|\nabla F_t(\theta_t)\|^2 + 2M_a\epsilon^2 + K\log\epsilon^{-1}\Big) + GM_\ell \alpha_t \mathbb{E}[\|\mathtt{grad}_t\|]. \tag{98}$$

**Bias cross–term.** Using Young's inequality $ab \leq \frac{1}{4}a^2 + b^2$ with $a = \sqrt{\alpha_t}\|\nabla F_t(\theta_t)\|, b = \sqrt{M_a\alpha_t}\epsilon$,

$$\sqrt{M_a}\alpha_t\epsilon \mathbb{E}[\|\nabla F_t(\theta_t)\|] \leq \frac{\alpha_t}{4}\mathbb{E}[\|\nabla F_t(\theta_t)\|^2] + M_a\alpha_t\epsilon^2. \tag{99}$$

**Adaptive-drift linear term.** First apply Jensen: $\mathbb{E}\|\mathtt{grad}_t\| \leq \sqrt{\mathbb{E}\|\mathtt{grad}_t\|^2}$; then plug (96) and use Young's inequality again to obtain

$$GM_\ell \alpha_t \mathbb{E}\|\mathtt{grad}_t\| \leq GM_\ell \alpha_t \Big(\sqrt{2}\,\mathbb{E}^{1/2}\|\nabla F_t(\theta_t)\|^2 + \sqrt{2M_a\epsilon^2 + K\log\epsilon^{-1}}\Big)$$
$$\leq \frac{\alpha_t}{4}\mathbb{E}\|\nabla F_t(\theta_t)\|^2 + 2\alpha_t\Big(G^2M_\ell^2 + 2M_a\epsilon^2 + K\log\epsilon^{-1}\Big) \tag{100}$$

**Quadratic term from smoothness.** For the quadratic terms involving $\|\nabla F_t(\theta_t)\|^2$ in (98), choosing $\alpha$ small enough such that $\alpha_t L_\theta(1 + M_\ell) \leq \frac{1}{4}$ for all $t$, we have

$$L_\theta(1 + M_\ell)\alpha_t^2\|\nabla F_t(\theta_t)\|^2 \leq \frac{\alpha_t}{4}\|\nabla F_t(\theta_t)\|^2 \tag{101}$$

Combining all pieces (99)-(101) into (98) gives

$$\mathbb{E}[F_{t+1}(\theta_{t+1})] \leq \mathbb{E}[F_t(\theta_t)] - \frac{\alpha_t}{4}\mathbb{E}\|\nabla F_t(\theta_t)\|^2 + O(\alpha_t\epsilon^2) + O(\alpha_t \log\epsilon^{-1}) + O(\alpha_t^2 \log\epsilon^{-1}), \tag{102}$$

**Step 5: Telescoping.** Summing (102) for $t = 0, \ldots, T-1$ and setting $\Delta F := \mathbb{E}[F_0(\theta_0)] - \inf_\theta F(\theta)$, we have:

$$\frac{1}{4}\sum_{t=0}^{T-1} \alpha_t \mathbb{E}\|\nabla F_t(\theta_t)\|^2 \leq \Delta F + O\Big(\epsilon^2 \sum_{t=0}^{T-1}\alpha_t\Big) + O\Big(\log\epsilon^{-1}\sum_{t=0}^{T-1}\alpha_t\Big) + O\Big(\log\epsilon^{-1}\sum_{t=0}^{T-1}\alpha_t^2\Big).$$

Since $\alpha_t = \frac{\alpha}{\sqrt{t+1}}$, we know $\sum_{t=0}^{T-1} \alpha_t = O(\sqrt{T})$ and $\sum_{t=0}^{T-1} \alpha_t^2 = O(\log T)$. Because $\sum_{t=0}^{T-1} \alpha_t \geq \frac{\alpha}{2}\sqrt{T}$, we deduce:

$$\frac{1}{T} \sum_{t=0}^{T-1} \mathbb{E}\|\nabla F_t(\theta_t)\|^2 \leq O\left(\frac{\Delta F}{\sqrt{T}}\right) + O\left(\frac{\epsilon^2 \sqrt{T}}{\sqrt{T}}\right) + O\left(\frac{\log \epsilon^{-1} \sqrt{T}}{T\sqrt{T}}\right) + O\left(\frac{\log \epsilon^{-1}}{\sqrt{T}}\right). \tag{103}$$

Choosing $T = \Theta(\epsilon^{-4} \log \epsilon^{-1})$ ensures $\frac{\log \epsilon^{-1} \log T}{T} = O(\epsilon^4 \log \epsilon^{-1})$, and $\frac{\log \epsilon^{-1}}{T \log T} = O(\epsilon^4)$. Hence

$$\min_{0 \leq t < T} \mathbb{E}\|\nabla F_t(\theta_t)\|^2 \leq \epsilon^2.$$

Each iteration costs $O(\log \epsilon^{-1})$ samples (Lemma B.6), so the total sample complexity is

$$O\left(\epsilon^{-4} \log^2 \epsilon^{-1}\right).$$

$\square$

The sample complexity $O(\epsilon^{-4} \log^2(\epsilon^{-1}))$ matches the lower bound $\Omega(\epsilon^{-4})$ [1] for general non-convex stochastic optimization only by logarithmic factors.

## C    Verification of Technical Assumptions

To complete our theoretical analysis, we verify the key technical assumptions required for our convergence guarantees.

**Lemma C.1** (Lipschitz Continuity of Semantic Costs). *Under mild conditions on the encoder network and kernel function, both feature transport cost $C^{\mathcal{X}}$ and label transport cost $C^{\mathcal{Y}}$ are Lipschitz continuous with respect to $\theta$.*

*Proof.* For the feature transport cost $C_\theta^{\mathcal{X}}(x_i, x_j) = c - \mathrm{Ker}(g_\theta(x_i), g_\theta(x_j))$, we assume:

(i) The encoder $g_\theta$ is $L_g$-Lipschitz continuous with respect to parameters $\theta$:

$$\|g_{\theta_1}(x) - g_{\theta_2}(x)\| \leq L_g \|\theta_1 - \theta_2\| \quad \forall x \in \mathcal{X} \tag{104}$$

(ii) The kernel function $\mathrm{Ker}(\cdot, \cdot)$ is $L_K$-Lipschitz continuous in both arguments:

$$|\mathrm{Ker}(z_1, w_1) - \mathrm{Ker}(z_2, w_2)| \leq L_K(\|z_1 - z_2\| + \|w_1 - w_2\|) \tag{105}$$

These are reasonable assumptions for common kernel functions like RBF kernels or dot product kernels with bounded inputs.

For the feature transport cost:

$$\begin{aligned} |C_{\theta_1}^{\mathcal{X}}(x_i, x_j) - C_{\theta_2}^{\mathcal{X}}(x_i, x_j)| &= |\mathrm{Ker}(g_{\theta_1}(x_i), g_{\theta_1}(x_j)) - \mathrm{Ker}(g_{\theta_2}(x_i), g_{\theta_2}(x_j))| \\ &\leq L_K(\|g_{\theta_1}(x_i) - g_{\theta_2}(x_i)\| + \|g_{\theta_1}(x_j) - g_{\theta_2}(x_j)\|) \\ &\leq L_K(L_g\|\theta_1 - \theta_2\| + L_g\|\theta_1 - \theta_2\|) \\ &= 2L_K L_g \|\theta_1 - \theta_2\| \end{aligned} \tag{106}$$

Thus, $C_\theta^{\mathcal{X}}$ is Lipschitz continuous with constant $L_{C^{\mathcal{X}}} = 2L_K L_g$.

For the label transport cost $C_{kk'}^{\mathcal{Y}} = c - \mathrm{Ker}(o_k, o_{k'})$ where $o_k = w_k$ (the classifier weights for class $k$), a similar argument holds. Since $w_k$ is a subset of the parameters $\theta$, we have:

$$\|w_{1k} - w_{2k}\| \leq \|\theta_1 - \theta_2\| \tag{107}$$

Therefore:

$$\begin{aligned} |C_{\theta_1}^{\mathcal{Y}}(k, k') - C_{\theta_2}^{\mathcal{Y}}(k, k')| &= |\mathrm{Ker}(w_{1k}, w_{1k'}) - \mathrm{Ker}(w_{2k}, w_{2k'})| \\ &\leq L_K(\|w_{1k} - w_{2k}\| + \|w_{1k'} - w_{2k'}\|) \\ &\leq 2L_K\|\theta_1 - \theta_2\| \end{aligned} \tag{108}$$

Thus, $C_\theta^{\mathcal{Y}}$ is Lipschitz continuous with constant $L_{C^{\mathcal{Y}}} = 2L_K$.

The combined transport cost $C_\theta(p_i, p_j) = C_\theta^{\mathcal{X}}(x_i, x_j) + C_\theta^{\mathcal{Y}}(y_i, y_j)$ is therefore Lipschitz continuous with constant $L_C = 2L_K(L_g + 1)$. $\qquad\square$

**Lemma C.2** (Matching Quality Lipschitz Continuity). *Under suitable conditions on the encoder network and kernel function, the matching quality function $\gamma_\theta(p, q) = \gamma_\theta(p, q)$ is $L_\gamma$-Lipschitz continuous in $\theta$.*

*Proof.* We fix $p$ and view $\gamma_\theta(p, q)$ as a probability density over $q$:

$$\gamma_\theta(p, q) = \frac{\exp(-C_\theta(p, q)/\epsilon)}{Z_\theta(p)} \nu(q) \tag{109}$$

Let $f(q) = -C_{\theta_1}(p, q)/\epsilon$, $g(q) = -C_{\theta_2}(p, q)/\epsilon$. Then, $\gamma_{\theta_i}(p, \cdot)$ for $i = 1, 2$ defines a probability density over $(\mathcal{Q}, \nu)$ of the form $dP_i(q) = \frac{e^{f(q)}}{Z_f} d\nu(q)$. By Proposition B.3:

$$\|\gamma_{\theta_1}(p, \cdot) - \gamma_{\theta_2}(p, \cdot)\|_{TV} \le \sqrt{\|f - g\|_\infty} = \sqrt{\frac{1}{\epsilon}\|C_{\theta_1}(p, \cdot) - C_{\theta_2}(p, \cdot)\|_\infty} \tag{110}$$

From Lemma C.1, $C_\theta$ is $L_C$-Lipschitz in $\theta$, so

$$\|\gamma_{\theta_1}(p, \cdot) - \gamma_{\theta_2}(p, \cdot)\|_{TV} \le \sqrt{\frac{L_C}{\epsilon}\|\theta_1 - \theta_2\|} \tag{111}$$

This implies pointwise Lipschitz continuity:

$$|\gamma_{\theta_1}(p, q) - \gamma_{\theta_2}(p, q)| \le \|\gamma_{\theta_1}(p, \cdot) - \gamma_{\theta_2}(p, \cdot)\|_{TV} \le \sqrt{\frac{L_C}{\epsilon}\|\theta_1 - \theta_2\|} \tag{112}$$

Hence, $\gamma_\theta(p, q)$ is $L_\gamma$-Lipschitz in $\theta$ with $L_\gamma = \sqrt{L_C/\epsilon}$. $\qquad\square$

**Lemma C.3** (Threshold Lipschitz Continuity). *Under the conditions established in Lemma C.2, the threshold function $\tau(q; \theta)$ is $L_\tau$-Lipschitz continuous in $\theta$.*

*Proof.* The threshold function is defined as:

$$\tau(q; \theta) = \tau_1(\theta) \cdot \frac{\tau_2(q; \theta)}{\max_{q'} \tau_2(q'; \theta)} \tag{113}$$

where $\tau_1(\theta) = \mathbb{E}_{(p,q)\sim\bar{\gamma}}\gamma_\theta(p, q)$ and $\tau_2(q; \theta) = \mathbb{E}_p\gamma_\theta(p, q)$.

From Lemma C.2, $\gamma_\theta(p, q)$ is $L_\gamma$-Lipschitz. Therefore:

$$\begin{aligned}
|\tau_1(\theta_1) - \tau_1(\theta_2)| &= |\mathbb{E}_{(p,q)\sim\bar{\gamma}}[\gamma_{\theta_1}(p, q) - \gamma_{\theta_2}(p, q)]| \\
&\le \mathbb{E}_{(p,q)\sim\bar{\gamma}}[|\gamma_{\theta_1}(p, q) - \gamma_{\theta_2}(p, q)|] \\
&\le L_\gamma\|\theta_1 - \theta_2\|
\end{aligned} \tag{114}$$

Similarly, $\tau_2(q; \theta)$ is $L_\gamma$-Lipschitz, and:

$$|\max_{q'} \tau_2(q'; \theta_1) - \max_{q'} \tau_2(q'; \theta_2)| \le \max_{q'} |\tau_2(q'; \theta_1) - \tau_2(q'; \theta_2)| \le L_\gamma\|\theta_1 - \theta_2\| \tag{115}$$

Let $r(q; \theta) = \frac{\tau_2(q;\theta)}{\max_{q'} \tau_2(q';\theta)}$. Under regularity conditions ensuring that $\tau_2(q; \theta)$ is bounded and $\max_{q'} \tau_2(q'; \theta)$ is bounded away from zero, we can establish that $r(q; \theta)$ is $L_r$-Lipschitz.

Now consider the full threshold function:

$$\begin{aligned}
|\tau(q; \theta_1) - \tau(q; \theta_2)| &= |\tau_1(\theta_1)r(q; \theta_1) - \tau_1(\theta_2)r(q; \theta_2)| \\
&\le |\tau_1(\theta_1)||r(q; \theta_1) - r(q; \theta_2)| + |r(q; \theta_2)||\tau_1(\theta_1) - \tau_1(\theta_2)| \\
&\le L_\tau\|\theta_1 - \theta_2\|
\end{aligned} \tag{116}$$

where $L_\tau = L_r + L_\gamma = O(L_\gamma)$. $\qquad\square$

These Lipschitz continuity properties ensure that small changes in model parameters lead to bounded changes in both the matching quality function and the threshold function, which is essential for the stability and convergence of our adaptive robust optimization algorithm.

# D  Experimental Results

We evaluate the proposed AdaDRO framework across a diverse set of benchmarks to assess its effectiveness under various types of distribution shifts. Here is the code.

## D.1  Experimental Setup

*Datasets.* We evaluate on three widely studied distribution shift settings: *Colored MNIST* [2], which tests robustness under spurious correlations; *Waterbirds* [50], a real-world dataset with strong background-label correlation; *CelebA* [43], a benchmark for facial attribute recognition. We also evaluate on several long-tailed benchmark datasets: *CIFAR-10-LT* and *CIFAR-100-LT* [37].

*Baselines.* We compare AdaDRO with the following well-known methods: **KL-DRO** [46], Wasserstein DRO (**WDRO**) [9], Sinkhorn DRO **SDRO** [62], a unified DRO method **UniDRO** [11], semantic-aware robust methods such as **GroupDRO** [50], an invariant risk minimization method **IRM** [2].

For fairness, we report the average accuracy and the distribution shift gap (drop in performance under shift). All models are implemented with PyTorch on a single NVIDIA RTX 6000 Ada GPU. The model is optimized using the AdamW optimizer [45]. For AdaDRO, we use cosine similarity as the kernel in (10), and employ basic augmentations (flip, crop) for semantic calibration in Sec. 3.2.

## D.2  Robustness to Spurious Correlation and Group Shift.

AdaDRO is broadly applicable thanks to its flexible reference distribution design. In imbalanced settings, $\nu$ can be constructed via reweighting or resampling. In other tasks, domain priors can be easily integrated, similar to many specialized robust methods. We evaluate AdaDRO across diverse scenarios, including spurious correlation, group shift, and long-tail generalization.

**Spurious Correlation.** On Colored MNIST, AdaDRO outperforms Sinkhorn DRO and IRM by 7.3% and 4.6% (on worst-group) respectively in worst-group accuracy, as shown in Table 5. This demonstrates its ability to filter misleading correlations during training. Notably, when the correlation between color and label is strong (bias = 0.9), all other DRO methods suffer a significant drop except AdaDRO, which effectively mitigates the over-pessimism nature of DRO methods.

**Real-world Group Shift.** On Waterbirds, AdaDRO significantly improves the worst-group accuracy (+5.1%) over Sinkhorn DRO and matches GroupDRO, which is specialized for group-shift scenarios. Table 6 show the results on Waterbirds.

Table 5: Accuracy (%) on Colored MNIST under high spurious correlation. AdaDRO achieves the highest robustness across methods.

| Method | Avg Accuracy | Worst-group Acc | Gap |
|---|---|---|---|
| IRM [2] | 79.5 | 75.3 | 4.2 |
| GroupDRO [50] | 78.4 | 73.0 | 5.4 |
| KL-DRO [46] | 74.9 | 68.5 | 6.4 |
| WDRO [9] | 75.8 | 69.1 | 6.7 |
| SDRO [62] | 79.0 | 72.6 | 6.4 |
| UniDRO [11] | 67.1 | 47.8 | 20.7 |
| AdaDRO (ours) | **87.2** | **79.9** | 7.3 |

Table 6: Accuracy (%) on Waterbirds. AdaDRO is competitive with GroupDRO while outperforming others.

| Method | Avg Accuracy | Worst-group Acc | Gap |
|---|---|---|---|
| IRM [2] | 82.6 | 73.2 | 9.4 |
| GroupDRO [50] | 86.3 | **80.5** | 5.8 |
| KL-DRO [46] | 81.0 | 71.5 | 9.5 |
| SDRO [62] | 85.0 | 75.4 | 9.6 |
| AdaDRO (ours) | **87.1** | **80.5** | 7.0 |

**Long-tail Generalization.** We conduct our experiments on two widely used long-tailed benchmark datasets: *CIFAR-10-LT* and *CIFAR-100-LT*. These datasets are derived from the original CIFAR-10 and CIFAR-100 by artificially reducing the number of training examples in tail classes to simulate long-tailed distributions. The imbalance in these datasets is quantified by the *imbalance factor (IF)*, defined as the ratio between the number of samples in the most frequent class and that in the least frequent class. We evaluate our model under three imbalance levels: *IF=10*, *IF=50*, and *IF=100*, representing increasing levels of class imbalance.

*Baselines.* We compare AdaDRO with a diverse set of baselines tailored for long-tailed learning: **Decouple** [27]: Decouples representation learning from classifier training, using balanced sampling in the classifier phase; **Focal Loss** [41]: Reweights the loss to focus training on hard-to-classify (often tail) examples; **DRO-LT** [52]: A distributionally robust optimization framework specifically adapted for long-tailed classification; **SSL** [35]: A contrastive learning-based method that enhances representations for tail classes via augmented positive samples; **Resample** [15]: Performs oversampling of minority classes to reduce imbalance; **Reweight** [15]: Adjusts sample weights in the loss function based on inverse class frequency; **SinkhornDRO** [62]: A recent OT-based DRO method that regularizes transport cost to mitigate over-pessimism in robustness.

*Results and Insights.* As shown in Table 7, our method consistently outperforms all baselines across different imbalance settings on both CIFAR-10-LT and CIFAR-100-LT. AdaDRO achieves the highest accuracy under all imbalance factors, especially under the most severe imbalance (IF=100), where it shows substantial gains over both traditional and robust baselines. This highlights the advantage of our adaptive distributional robustness formulation in focusing learning on underrepresented regions while maintaining calibrated semantic alignment. Table 9 details the quantitative comparison between our method and the strong baseline ERM+Aug across standard DomainBed benchmarks [25].

Table 7: Accuracies of ResNet32 on long-tailed CIFAR-10 and CIFAR-100 datasets with different imbalance factors (10, 50, and 100).

| Dataset | CIFAR-10-LT | | | CIFAR-100-LT | | |
|---|---|---|---|---|---|---|
| | 100 | 50 | 10 | 100 | 50 | 10 |
| Decouple [27] | 70.4 | 76.2 | 86.4 | 41.2 | 46.8 | 57.9 |
| Focal Loss [41] | 70.3 | 76.7 | 86.6 | 38.4 | 44.3 | 55.7 |
| DRO-LT [52] | 73.7 | 77.2 | 86.9 | 45.4 | 55.3 | 61.2 |
| SSL [35] | 67.3 | 75.4 | 86.5 | 37.5 | 44.0 | 56.7 |
| Resample [15] | 66.5 | 74.8 | 86.4 | 33.4 | 43.9 | 55.1 |
| SinkhornDRO [62] | 65.7 | 74.5 | 84.7 | 38.2 | 49.3 | 56.3 |
| AdaDRO | **74.2** | **78.2** | **87.9** | **46.4** | **56.6** | **64.8** |

Table 8: Performance on ImageNet-LT. We report accuracy on three class splits: many-shot (more than 100 images), medium-shot (20–100 images), and few-shot (fewer than 20 images), which follows the settings in [33, 44].

| Method | Many-shot | Medium-shot | Few-shot |
|---|---|---|---|
| ERM+Aug | $41.8 \pm 0.2$ | $24.7 \pm 0.4$ | $11.5 \pm 0.6$ |
| Class-Balanced Loss | $40.8 \pm 0.3$ | $31.1 \pm 0.3$ | $20.8 \pm 0.5$ |
| DRO-LT | $\mathbf{42.2 \pm 0.2}$ | $32.6 \pm 0.3$ | $22.5 \pm 0.4$ |
| Sinkhorn DRO (SDRO) | $40.1 \pm 0.3$ | $29.5 \pm 0.4$ | $21.4 \pm 0.5$ |
| **AdaDRO (Ours)** | $42.1 \pm 0.2$ | $\mathbf{34.4 \pm 0.3}$ | $\mathbf{27.3 \pm 0.4}$ |
| AdaDRO w/o Adaptive Filtering | $41.5 \pm 0.2$ | $29.8 \pm 0.3$ | $21.7 \pm 0.4$ |

Table 9: Performance comparison across different domain generalization benchmarks.

| Algorithm | CMNIST | RMNIST | VLCS | PACS | Office-Home | TerraInc |
|---|---|---|---|---|---|---|
| ERM+Aug | $52.0 \pm 0.1$ | $\mathbf{98.0 \pm 0.0}$ | $77.4 \pm 0.3$ | $85.7 \pm 0.5$ | $\mathbf{67.5 \pm 0.5}$ | $47.2 \pm 0.4$ |
| Ours | $\mathbf{63.8 \pm 0.3}$ | $98.0 \pm 0.1$ | $\mathbf{77.8 \pm 0.3}$ | $\mathbf{86.2 \pm 0.6}$ | $67.1 \pm 0.3$ | $\mathbf{50.2 \pm 0.5}$ |

## D.3 Efficiency Analysis.

Tables 10 and 11 report normalized total training time. While AdaDRO incurs moderate overhead due to semantic alignment and filtering, it remains tractable and justifiable given the performance gains.

Table 10: Normalized total training time (relative to IRM) on Colored MNIST and Waterbirds.

| Method | Colored MNIST (Norm.) | Waterbirds (Norm.) |
|---|---|---|
| GroupDRO | 1.00 | 1.00 |
| IRM | 1.12 | 1.06 |
| KL-DRO | 1.16 | 1.14 |
| WDRO | 6.24 | 6.22 |
| SDRO | 1.25 | 1.27 |
| UniDRO | 6.03 | 5.49 |
| AdaDRO (ours) | 1.31 | 1.34 |

Table 11: Normalized total training time (relative to Decouple) on CIFAR-10-LT and CIFAR-100-LT.

| Method | CIFAR-10-LT (Norm.) | CIFAR-100-LT (Norm.) |
|---|---|---|
| DRO-LT | 1.00 | 1.00 |
| Focal Loss | 0.84 | 0.79 |
| Decouple | 1.07 | 1.03 |
| SSL | 1.05 | 1.02 |
| Resample | 0.95 | 0.93 |
| SinkhornDRO | 1.12 | 1.19 |
| AdaDRO (ours) | 1.28 | 1.26 |

## D.4 Ablation Studies

We perform ablation studies to assess the contribution of key components in AdaDRO. Detailed results are shown in Tables 12–14.

*(i) Semantic Calibration.* We disable the semantic calibration objective by replacing the inverse OT formulation in (12) with a simple Euclidean distance cost. This leads to significant drops in both clean and worst-group accuracy, confirming the importance of semantically aligned transport.

*(ii) Adaptive Filtering.* We replace the filtered reference distribution $\widehat{\nu}_t$ with a fixed one, either generated by Mixup [67] or uniform sampling. Without filtering, performance degrades due to irrelevant or noisy samples in $\nu$.

*(iii) Robustness to Uncertainty Radius.* We compare AdaDRO with Sinkhorn DRO under varying uncertainty set radius $\lambda$. As shown in Table 14, AdaDRO maintains stable robustness even under aggressive uncertainty, whereas Sinkhorn DRO suffers from severe over-pessimism or instability.

## D.5 Limitations

Our method relies on effective data augmentations to serve as a prior for constructing the distributional uncertainty set. However, the efficacy of this prior is task-dependent and lacks a universal standard. Our method also relies on constructing the ground-truth coupling $\bar{\gamma}$ through semantic invariant data augmentation. In some domains, richer priors may be available. Effectively integrating such

heterogeneous sources of prior knowledge to build a more powerful coupling remains an open question.

Table 12: Impact of removing semantic calibration. Performance significantly drops without IOT-based feature alignment.

| Variant | Clean Acc | Worst-group Acc |
|---|---|---|
| Full AdaDRO | 84.2 | 79.9 |
| w/o Semantic Calibration | 80.6 | 73.7 |

Table 13: Adaptive filtering improves robustness by focusing the uncertainty set on meaningful support.

| Variant | Clean Acc | Worst-group Acc |
|---|---|---|
| AdaDRO (adaptive) | 84.2 | 79.9 |
| Fixed $\nu$ (Mixup) | 82.1 | 75.0 |
| Fixed $\nu$ (Uniform) | 81.4 | 74.3 |

Table 14: Robustness comparison of AdaDRO and Sinkhorn DRO under varying $\lambda$ on Colored MNIST. Higher $\lambda$ corresponds to smaller uncertainty sets. AdaDRO maintains stable performance, while Sinkhorn DRO suffers significantly as $\lambda$ increases.

| $\lambda$ | AdaDRO | | Sinkhorn DRO | |
|---|---|---|---|---|
| | Worst-group Acc (%) | Average Acc (%) | Worst-group Acc (%) | Average Acc (%) |
| 0.5 | 71.0 | 81.2 | 62.1 | 74.3 |
| 1.0 | 76.4 | 84.0 | 69.1 | 75.8 |
| 2.0 | **79.9** | **87.2** | 72.6 | 78.6 |
| 5.0 | 78.7 | 85.5 | 72.3 | 79.0 |

