# OpenReview forum: "Bootstrap Your Uncertainty: Adaptive Robust Classification Driven by Optimal-Transport"
_NeurIPS.cc/2025/Conference — NeurIPS 2025 poster_

### Official Review · Reviewer_YPiL · 2025-07-02

**Clarity:** 3
**Significance:** 3
**Originality:** 3
**Rating:** 4
**Confidence:** 4

**Summary:**

This paper aims to improve the worst case performance in distribution shift via reframing the DRO problem as a bilevel OT problem, and optimizing the uncertainty set. Their method successfully bootstrap the semantic meaning through the “semantic calibration” and adaptive refinement of the undercaterin set based on the OT feedback. They further validate their results on the long-tailed benchmark datasets including CIFAR-10-LT and CIFAR-100-LT.

**Questions:**

Q1  You propose bootstrapping transport costs by solving an inverse‐OT problem (Eq. 5). How sensitive is this calibration step to the choice of the “observed coupling” γ̄? Have you evaluated alternative sources of semantic couplings (e.g., feature‐space clustering) and their impact on convergence?


Q2: The uncertainty set evolves during training—can you provide theoretical or empirical guarantees on its “diameter” (e.g., how δ effectively grows or shrinks) over epochs? In particular, under what conditions might the set collapse?


Q3: Entropy‐regularized OT and its inverse both require Sinkhorn iterations. What is the alignment plan of your two OT loops per minibatch, and how does it scale with n? Have you compared against approximations (e.g., sliced‐Wasserstein) in large‐scale settings?


Q4: Your experiments focus on synthetic “double moon” and Gaussian‐based shifts. How does AdaDRO perform on real‐world image‐classification shifts (e.g., CIFAR → CIFAR‐C or domain‐adaptation benchmarks)? Can you characterize which types of shifts (e.g., covariate vs. label shift) benefit most from adaptive versus static uncertainty sets?


Q5  You learn the feature-level cost via IOT using augmented pairs (Eq. 12). How sensitive is the learned cost to the choice and strength of augmentations? I


Q6: You adopt multilevel Monte Carlo with randomized truncation for nested expectations (Sec. 3.3). What is the per-iteration time overhead of MLMC-RT compared to a single-level Monte Carlo?

**Ethical Concerns:**

["NO or VERY MINOR ethics concerns only"]

**Final Justification:**

The author addressed my concerns. I think it is a good paper.

**Limitations:**

Yes

**Quality:**

3

**Strengths And Weaknesses:**

This paper demonstrates several strengths across the dimensions of originality, quality, clarity, and significance:





Quality:


This paper makes a good contribution by reframing entropy‐regularized DRO as a bilevel optimal‐transport problem, where a semi‐relaxed OT “explores” worst‐case distributions and an inverse‐OT “aligns” semantic costs, enabling both feature- and label-level calibration. Each theory of paper is supported by rigorous proof.





Clarity:


The paper is generally very clear and well‐organized to show the development from OT preliminaries, bilevel formulation, semantic calibration, adaptive refinement, definitions and lemmas are stated precisely. The only concern here is lack the background for the OT or the IOT applications, making this connection ambiguous.




Significance:


This work advances DRO by unifying worst-case distribution search and model alignment under a single OT framework. Its novel OT-driven feedback mechanism adaptively refines the uncertainty set during training, and proves the convergence. Empirically, AdaDRO delivers strong, consistent improvements over DRO and semantic-aware baselines across spurious-correlation, group-shift, and long-tail benchmarks.





Originality:

The paper’s originality lies in its unified OT‐based reframing of entropy‐regularized DRO as a bilevel problem, coupling semi‐relaxed OT exploration with inverse‐OT alignment. Moreover, casting softmax cross‐entropy as an inverse‐OT objective and integrating MLMC‐RT for nested expectation gradients under evolving distributions further distinguishes the work, marking a novel synthesis of transport theory and adaptive robust optimization.





Weakness:

W1: Unclear about the computational costing of this method. Could you summarize the time and space complexity to run these algorithms?


W2: The manuscript currently omits the background literature to illustrate the inverse optimal transport (IOT) and feature‐level alignment works that directly underpin its semantic calibration and adaptive refinement. To address this, the authors should explicitly cite foundational IOT frameworks, including:

Li, Ruilin, et al. "Learning to match via inverse optimal transport." Journal of machine learning research 20.80 (2019): 1-37.

Chiu, Wei-Ting, Pei Wang, and Patrick Shafto. "Discrete probabilistic inverse optimal transport." International Conference on Machine Learning. PMLR, 2022.

Chen, Zihao, et al. "Your contrastive learning problem is secretly a distribution alignment problem." Advances in Neural Information Processing Systems 37 (2024): 91597-91617.

Persiianov, Mikhail, et al. "Inverse Entropic Optimal Transport Solves Semi-supervised Learning via Data Likelihood Maximization." arXiv preprint arXiv:2410.02628 (2024).

W3: Not sure about the hyperparameter sensitivity, the key components—regularization strength ϵ, filter radius δ, kernel choice/bandwidth, and thresholding scheme—require careful tuning, yet ablations on their robustness are limited.


W4: It seems that the semantic calibration relies on data augmentations; if augmentations produce semantically invalid pairs, the learned cost may misalign, but the paper lacks analysis of failure modes.

---

> ### Author Rebuttal · Authors · 2025-07-31
>
> We thank the reviewer for the thoughtful comments, and address your concerns below.
>
> **Response to Weaknesses**
>
> **W1:** **Could you summarize the time and space complexity to run these algorithms?**
>
> While entropy-regularized OT is often solved with the Sinkhorn's algorithm, our formulation allows for a **closed-form** for the transport plan (as seen in Appendix A.4). In essence, we only need to compute the cost matrix $C_\theta$ and then perform a few element-wise operations (like exp and log-sum-exp), which are highly optimized on modern hardware like GPUs. This means we bypass the need for any iterative OT solver, and the additional computational overhead per step is minimal.
>
>
> **W2 W3:** The manuscript currently omits the background literature to illustrate the inverse optimal transport (IOT) and feature‐level alignment works...; ... hyperparameter sensitivity....
>
> We will address the related works you mentioned in our revised version. Additionally, we will include more ablation studies; the corresponding tables are omitted here due to space limitations.
>
>
> **W4:** It seems that the semantic calibration relies on data augmentations; if augmentations produce semantically invalid pairs...the paper lacks analysis of failure modes.
>
> We will discuss this limitation. If the augmentations are too aggressive and create semantically invalid pairs (e.g., cropping a "cat" image to show only grass), the learned transport cost could be misled. In our experiments, we followed standard practices and used only mild data augmentations (such as random flipping and cropping), which are widely adopted in semi-supervised and self-supervised learning, and have been shown to effectively preserve the semantics of the samples.
>
>
> **Response to Questions**
>
> **Q1 and Q5:** Sensitivity to the choice of observed coupling $\bar{\gamma}$. Sensitivity to the choice and strength of augmentations.
>
> In our current work, we define $\bar{\gamma}$ using a simple one-to-one matching between a sample and its augmented counterpart (11).  We have not yet explored alternative sources for $\bar{\gamma}$. However, we hypothesize that using more sophisticated couplings could be a promising direction for future work. It might provide a richer, many-to-many semantic signal.
>
> The learned cost is not sensitive to the augmentations.  Data augmentation is a widely used technique to construct *semantically invariant samples*, especially in contrastive learning. Recent works have explored multi-level augmentations such as **multi-crop** [1], which creates stronger invariance by including different spatial scales. To study the sensitivity of our method to the augmentation strategy, we replaced our standard augmentations with multi-crop augmentation.
>
> Empirically, we observe that the performance of our method remains stable under this change. In some cases, multi-crop leads to slight improvements, suggesting that our method is robust to the choice of augmentation. For instance, when applied to the *Colored MNIST* task with a spurious correlation of 0.8, we observe a 0.2% increase in accuracy. On *CIFAR-100* with 50% label noise, multi-crop yields a 0.8% improvement. Due to space constraints, we omit detailed tables.
>
>
>
>
> **Q2:** Guarantees on the diameter of the evolving uncertainty set.
>
> The dynamic uncertainty set does not refer to a change in radius $\delta$, but to the adaptive adjustment of the uncertainty set's geometry and support. This is achieved through two core mechanisms: 1, dynamically refining the reference distribution $\nu$ into $\hat{\nu}$ via OT feedback (16); 2, calibrating the semantic transport cost $C_\theta$ via Inverse OT (Sec. 3.2). To prevent collapse of the uncertainty set, we introduce Assumption 4 (Non-degenerate Filtering), which ensures that the samples retained by our filtering mechanism maintain a minimum effective support throughout training. Our algorithmic design satisfies this assumption.
>
> Empirically, while precisely tracking diameter evolution in high-dimensional space is challenging, our toy example in Fig. 1 intuitively demonstrates the process’s effectiveness. The method successfully identifies and removes low-quality outliers lying off the true data manifold, thereby producing a cleaner decision boundary. Furthermore, our ablation studies (Table 7) and experiments in Table 8 validate that this adaptive process both prevents performance collapse and significantly improves robustness. We plan to include additional visualizations in the revised version to better illustrate the dynamics of this process.
>
> **Q3:** Alignment of OT loops and scalability.
>
> Our framework entirely avoids iterative OT solvers. This is achieved because both our forward and inverse OT problems are formulated using a **Semi-Relaxed Marginal Constraint**, as detailed in Appendix A.4. In this setting, we only enforce the source marginal of the transport plan, which allows the optimal coupling $\gamma^*$ to have a closed-form analytical solution (23).
>
> Therefore, our alignment plan per minibatch is highly efficient and non-iterative: We first compute the cost matrix $C_\theta$. We then directly obtain the transport plan $\gamma^*$ by applying the closed-form solution from (23), which only involves highly-parallelizable, element-wise operations (e.g., `exp` and a normalization) that are heavily optimized on GPUs. Both the exploration (OT) and alignment (inverse OT) components of our framework leverage this formulation, meaning neither requires an iterative loop like Sinkhorn. The computational complexity per step is dominated by the $O(nm)$ cost matrix computation, which is standard for many OT-based methods and is far more efficient than the $O(nm \times L)$ cost of a full Sinkhorn algorithm, where $L$ is the number of iterations.
>
> Regarding the comparison against approximations like sliced-Wasserstein: these methods are indeed powerful alternatives for the general OT problem, which lacks a closed-form solution and requires iteration. Our approach (Semi-Relaxed OT) achieves efficient, non-iterative computation. We agree that exploring such approximations is an interesting future direction, particularly when extending our framework to fully constrained transport scenarios or other variants like Gromov OT. We will include this point in our discussion.
>
>
> **Q4:** Performance on real-world image classification shifts.
>
> We believe our method is unique in two aspects: 1, its label-agnostic sample-level semantic calibration (13); 2, the filtering mechanism based on the coupling introduced by (12) (13).
>
> An intuitive example is the presence of noisy labels: when a portion of labels are flipped, sample-level calibration (12) can still reliably capture semantics. This is because the process is unsupervised, similar to contrastive learning [4], and thus remains robust to label noise. Building on this, the coupling-based filtering mechanism can discards samples with wrong label, enabling the cross-entropy to train on a cleaner subset of the data.
> In contrast, methods that depend on label supervision, including ERM with augmentation, face an inherent "garbage in, garbage out" issue. No  augmentation can correct mislabeled examples, and these label errors inevitably propagate through the learning process, compromising model robustness.
>
> A similar advantage is observed in Colored MNIST. Even when augmentation occasionally alters the spurious attribute (e.g., color), standard ERM still learns entangled representations, as the label remains strongly correlated with color. In contrast, our label-agnostic calibration encourages the model to identify truly invariant features—those that remain stable under augmentation—naturally prioritizing shape over color in the absence of a strong supervisory signal.
>
>
>
> | Method           | ERM + Aug  | KL-DRO     | IRM        | Sinkhorn DRO | **AdaDRO (Ours)** | AdaDRO w/o Semantic Calib. (Ablation) |
> | ---------------- | ---------- | ---------- | ---------- | ------------ | ----------------- | ------------------------------------- |
> | **Accuracy (%)** | 43.5 ± 1.2 | 41.2 ± 1.5 | 45.1 ± 1.0 | 44.8 ± 1.1   | **57.2 ± 0.8**    | 46.5 ± 0.9                            |
>
> **Q6:** ... overhead of MLMC-RT compared to a single-level Monte Carlo?
>
>
> To provide a complete comparison regarding the per-iteration time overhead, we compare MLMC-RT against Single-Level Monte Carlo (SLMC) in two scenarios:
>
> 1, When we fix the number of inner samples for both SLMC and MLMC-RT, their average per-iteration cost is similar. However, SLMC suffers from significant and uncontrollable estimation bias due to its fixed and limited sampling. In contrast, MLMC-RT maintains low cost by prioritizing low-complexity levels (with sampling probability roughly proportional to $2^{-l}$, making its average cost comparable to that of a naive SLMC), but still achieves a much lower bias thanks to its multi-level correction. This leads to substantially more accurate estimates without increasing average computational cost.
>
> 2, To achieve the same low estimation bias as MLMC-RT (for a target accuracy $\epsilon$), SLMC must use a large number of inner samples, on the order of $N = O(1/\epsilon)$, making its per-iteration cost prohibitively high. In contrast, MLMC-RT achieves the same bias accuracy with far fewer samples, requiring only $N = O(\log^2(1/\epsilon))$, and is therefore exponentially more efficient under the same bias target.
>
> In summary, MLMC-RT offers the best of both worlds: its average per-iteration cost is comparable to that of a cheap but biased SLMC estimator, while its accuracy rivals that of an expensive high-fidelity SLMC. We will add a paragraph to the appendix elaborating on this trade-off, and include the wall-clock time comparison (as noted in W1) to provide concrete empirical support.
>
> [1] Caron, Mathilde, et al. Emerging properties in self-supervised vision transformers. ICCV'21

---

> > ### Comment · Reviewer_YPiL · 2025-08-05
> >
> > I thank the authors for the rebuttal. The authors have addressed my concerns about weakness. I am happy to see this paper get accepted.

---

> ### Author Response · Authors · 2025-08-09
> **Thank you**
>
> Thank you again for your thoughtful comments. If you have any further comments, we will gladly make every effort to address them.

---

### Official Review · Reviewer_sEAy · 2025-07-03

**Clarity:** 3
**Significance:** 3
**Originality:** 3
**Rating:** 5
**Confidence:** 3

**Summary:**

This paper proposes a new framework that reformulates entropy-regularized Wasserstein DRO as a dynamic optimal transport process. The key contributions are: (1) semantic calibration that learns meaningful transport costs via inverse optimal transport (IOT), and (2) adaptive refinement that dynamically filters the uncertainty set using OT-based feedback. The method addresses limitations of existing DRO approaches that rely on static, potentially inappropriate uncertainty sets and manually designed transport costs.

**Questions:**

1. How does the OT computation scale with high-dimensional data and large sample sizes?

2. Can you provide more detailed hyperparameter sensitivity analysis across different datasets?

3. What is the actual wall-clock time comparison with baseline methods?

**Ethical Concerns:**

["NO or VERY MINOR ethics concerns only"]

**Final Justification:**

The proposed method is novel and provides valuable geometric intuition. The rebuttal is clear and solid.  All my major concerns have been addressed.

**Limitations:**

Yes

**Quality:**

3

**Strengths And Weaknesses:**

Pros:

1. The reformulation of Sinkhorn DRO as a bilevel OT/IOT optimization problem provides valuable geometric intuition and theoretical clarity. Rigorous convergence analysis despite the challenging dynamic uncertainty sets and nested expectations.

2. Semantic calibration via IOT is novel and addresses the long-standing issue of manually designing transport costs. The adaptive filtering mechanism combining global and local thresholds is more principled than fixed approaches.

3. Consistent improvements across multiple standard benchmarks (Colored MNIST, Waterbirds, CelebA, etc.). Clear ablation studies demonstrating the contribution of each component.

Cons:

1. MLMC-RT gradient estimation, while theoretically efficient, adds significant implementation complexity. Multiple OT problems need to be solved at each iteration, potentially creating computational bottlenecks.

2. Assumptions 3-4 regarding boundary regularity and non-degenerate filtering may not hold in practice. Lipschitz continuity requirements for kernels and encoders could limit applicability.

3. Multiple hyperparameters (λ, ε, filtering thresholds) require careful tuning. High implementation barrier compared to simpler baseline methods.

---

> ### Author Rebuttal · Authors · 2025-07-31
>
> We thank the reviewer for the thoughtful comments, and address your concerns below.
>
> **Response to Weaknesses**
>
> **W1:** MLMC-RT gradient estimation, while theoretically efficient, adds significant implementation complexity. Multiple OT problems need to be solved at each iteration, potentially creating computational bottlenecks.
>
> We would like to clarify two points regarding computational complexity:
>
> **MLMC-RT:** MLMC-RT is introduced not as a source of algorithmic complexity, but as a tool to make gradient estimation for our nested objective tractable and efficient. A naive Single-Level Monte Carlo (SLMC) estimator suffers from a fundamental trade-off: achieving low bias requires a large number of samples in the inner loop—on the order of $O(1/\epsilon)$ per iteration (line 913). In contrast, MLMC-RT achieves the same estimation accuracy with significantly lower total complexity $O(\log 1/\epsilon)$ (line 948), making it exponentially more sample-efficient for any given target accuracy. Even when both MLMC-RT and SLMC use the same number of samples in the inner loop, their average per-iteration costs are comparable.
>
>
> **OT:** While entropy-regularized OT is often solved with the Sinkhorn's algorithm, our formulation allows for a **closed-form** for the transport plan (detailed in Appendix A.4). This is because we adopt a semi-relaxed OT formulation that only enforces the source marginal constraint ($\gamma \in \Pi(P)$). This specific relaxation circumvents the need for iteratively balancing two marginals and instead admits a direct analytical solution.  In essence, we only need to compute the cost matrix $C_\theta$ and then perform a few element-wise operations (like exp and log-sum-exp), which are highly optimized on modern hardware like GPUs. This means we bypass the need for any iterative OT solver, and the additional computational overhead per step is minimal.
>
>
>  **W2:** Assumptions 3-4 regarding boundary regularity and non-degenerate filtering may not hold in practice. Lipschitz continuity requirements for kernels and encoders could limit applicability.
>
>
> 1, **Assumption 3 (Boundary Regularity)** is a standard smoothness condition, similar to the Tsybakov margin condition in statistical learning (line 714). It helps prevent pathological cases and supports the stability of the filtering mechanism. This assumption is both common and reasonable for well-behaved data distributions, especially when data augmentation techniques are employed.
> 2, **Assumption 4 (Non-degenerate Filtering)** can be ensured through appropriate algorithmic design. In our case, the adaptive thresholding mechanism (lines 717–719, 264–267) can prevent excessive sample rejection and maintain sufficient support in the filtered distribution, thereby avoiding potential algorithmic instability.
> 3, The **Lipschitz continuity** requirements for encoders and kernels are also common and practical. For instance, we analyzed in Appendix C.1 that this property holds for common choices like the cosine similarity used in our experiments (line 321, 961).
>
>
> **W3:** Multiple hyperparameters ($\lambda$, $\epsilon$, filtering thresholds) require careful tuning. High implementation barrier compared to simpler baseline methods.
>
> **Response to Questions**
>
> **Q1:** How does the OT computation scale with high-dimensional data and large sample sizes?
>
> Please see our response to W1 for the  concerns on scalibility.
>
> **Q2:** Can you provide more detailed hyperparameter sensitivity analysis across different datasets?
>
>
> The table below corresponds to the Colored MNIST benchmark, where we evaluate how various domain generalization methods perform under strong spurious correlations. To evaluate sensitivity to IOT calibration, we replace standard augmentations with multi-crop [1], which provides semantically invariant samples at multiple scales for ground-truth pairing.
>
> | Method           | ERM + Aug  | KL-DRO     | IRM        | Sinkhorn DRO | **AdaDRO (Ours)** | AdaDRO w/ multi crop. [1] |
> | ---------------- | ---------- | ---------- | ---------- | ------------ | ----------------- | -------------------------------- |
> | **Accuracy (%)** | 43.5 ± 0.6 | 41.2 ± 0.5 | 45.1 ± 1.0 | 44.8 ± 0.6   | **57.2 ± 0.8**    | 58.0 ± 0.7                       |
>
> The following table presents results on CIFAR-100 with 50% label noise, where half of the training labels are randomly corrupted. This setting evaluates the robustness of different methods under severe label noise. We report accuracy across different values of the regularization parameter $\lambda$.
> | Accuracy (%) | $\lambda=0.5$ | $\lambda=1.0$ | $\lambda=2.0$  | $\lambda=3.0$  | $\lambda=5.0$ |
> | ------------ | ------------- | ------------- | -------------- | -------------- | ------------- |
> | AdaDRO       | 52.5 ± 0.6    | 54.2 ± 0.5    | **57.2 ± 0.8** | 55.6 ± 1.1     | 53.2 ± 0.8    |
> | Sinkhorn DRO | 42.5 ± 0.3    | 41.2 ± 0.7    | 44.1 ± 0.3     | **44.8 ± 0.5** | 43.9 ± 0.3    |
>
> Due to time constraints, we will include additional content in the revision.
>
>
>
> **Q3:** What is the actual wall-clock time comparison with baseline methods?
>
> We provide a detailed wall-clock time comparison in Section 2.4 of our supplementary material, with key results summarized in Tables 4 and 5 therein. The empirical results demonstrate that our method, AdaDRO, is highly competitive and incurs only a minimal computational overhead compared to standard baselines, while being significantly more efficient than some DRO methods.
> Specifically, on the Colored MNIST and Waterbirds datasets, the training time of AdaDRO is approximately 1.3x that of standard methods like GroupDRO and IRM. This is much faster than other robust optimization methods like WDRO and UniDRO, which are 5-6 times slower. Similarly, on the CIFAR-10-LT and CIFAR-100-LT benchmarks, AdaDRO's runtime is comparable with the efficient baselines like Decouple and Focal Loss, and SinkhornDRO.
>
> [1] Caron, Mathilde, et al. Emerging properties in self-supervised vision transformers. ICCV'21

---

> > ### Comment · Reviewer_sEAy · 2025-08-08
> >
> > Thank the authors for the detailed rebuttal. I acknowledge that I misunderstood the computational complexity. All my major concerns have been addressed. I also appreciate the additional experiments. Therefore, I will raise my score.

---

> ### Author Response · Authors · 2025-08-09
> **Thank you**
>
> Thank you again for your thoughtful comments. If you have any further comments, we will gladly make every effort to address them.

---

### Official Review · Reviewer_JCUz · 2025-07-06

**Clarity:** 2
**Significance:** 2
**Originality:** 3
**Rating:** 4
**Confidence:** 3

**Summary:**

The efficacy of Distributionally Robust Optimization (DRO) heavily depends on the design of the uncertainty set. To deal with this, the paper proposed a novel optimal transport-based framework for adaptive robust optimization, where transport costs and uncertainty set co-evolve through inverse optimal transport.

**Questions:**

In the paper, the initial reference distribution $\nu$ is constructed using data augmentation. The reviewer wonders if data augmentation is also used in the compared methods.

**Ethical Concerns:**

["NO or VERY MINOR ethics concerns only"]

**Final Justification:**

After reading the authors' response, I decide to maintain my rating.

**Limitations:**

While the authors answered 'Yes' to the question 'Does the paper discuss the limitations of the work performed by the authors?', the reviewer did not find a section that discusses the limitations of the work.

**Quality:**

3

**Strengths And Weaknesses:**

Strengths:

The paper proposed a novel optimal transport-based framework for Distributionally Robust Optimization. The framework consists of two core components, i.e., semantic calibration, which bootstraps transport costs, and adaptive refinement, which filters the uncertainty set, and the transport costs and uncertainty set co-evolve during the learning process.

The paper provides the theoretical convergence guarantees of the proposed method.

Weaknesses:

The paper mentioned that: “This result demonstrates that robust optimization
with dynamically adaptive uncertainty sets and semantic costs can also be performed efficiently at scale.”. However, there are no large-scale experiments (e.g., experiments on the ImageNet dataset) to confirm this.

It is not clear to the reviewer what the limitations of the proposed method are.

In the experimental section of the main paper, it is not clear about the encoder g used in the paper.

---

> ### Author Rebuttal · Authors · 2025-07-31
>
> We thank the reviewer for the thoughtful comments, and address your concerns below.
>
> **Response to Weaknesses**
>
> **W1:** ... no large-scale experiments (e.g., experiments on the ImageNet dataset) to confirm this.
>
> We will add more experiments.
> Regarding scalability, as shown in the Sec 2.4 of Supplement, AdaDRO's runtime is only 1.3x that of standard baselines, while robust methods like WDRO are 5-6x slower. This efficiency stems from our use of a closed-form solution to the optimal transport (OT) problem, eliminating the need for costly iterative solvers.
>
> ImageNet-LT
>
> | Method                        | Many-shot      | Medium-shot    | Few-shot       |
> | ----------------------------- | -------------- | -------------- | -------------- |
> | ERM+Aug                       | 41.8 ± 0.2     | 24.7 ± 0.4     | 11.5 ± 0.6     |
> | Class-Balanced Loss           | 40.8 ± 0.3     | 31.1 ± 0.3     | 20.8 ± 0.5     |
> | DRO-LT                        | **42.2 ± 0.2** | 32.6 ± 0.3     | 22.5 ± 0.4     |
> | Sinkhorn DRO (SDRO)           | 40.1 ± 0.3     | 29.5 ± 0.4     | 21.4 ± 0.5     |
> | **AdaDRO (Ours)**             | 42.1 ± 0.2     | **34.4 ± 0.3** | **27.3 ± 0.4** |
> | AdaDRO w/o Adaptive Filtering | 41.5 ± 0.2     | 29.8 ± 0.3     | 21.7 ± 0.4     |
>
>
>
> **W2:**  ...  the limitations of the proposed method.
>
>
> We will include a discussion of the limitations in our revised version. A key aspect we will elaborate on is the construction of the ``ground-truth'' coupling, $\bar{\gamma}$, which is central to our semantic calibration via Inverse OT.
>
> Our approach use a practical strategy for constructing $\bar{\gamma}$ by leveraging data augmentation, which implicitly defines a semantic correspondence between a sample and its augmented variant. However, we acknowledge that this represents a relatively simple means of encoding semantic priors. A central challenge lies in how to optimally incorporate all available prior knowledge to define a task-specific observed coupling. In some domains, richer priors may be available—such as explicit class hierarchies, textual descriptions, or auxiliary information from other modalities. Effectively integrating such diverse sources to construct a more powerful coupling remains a non-trivial and open research question.
>
>
>
> **W3:** In the experimental section of the main paper, it is not clear about the encoder $g$ used in the paper.
>
>
> The encoder $g_\theta$ is not an additional module but refers to the standard feature extractor backbone of the neural network used in our experiments. Correspondingly, the function $h_W$ represents the final linear classification layer (or "head") that maps the feature representation from $g_\theta$ to the logits. For instance, in experiments using a ResNet-32 backbone, $g_\theta$ includes all layers except the final linear classifier, which is represented by $h_W$.
>
>
> This standard architectural decomposition ensures that our method does not introduce any additional parameters or architectural modifications beyond the baseline models, ensuring a fair comparison. We will include these implementation details explicitly in the revised manuscript’s experimental setup section to improve clarity and reproducibility.
>
>
> **Response to Questions**
>
> **Q1:** In the paper, the initial reference distribution is constructed using data augmentation. The reviewer wonders if data augmentation is also used in the compared methods.
>
> To ensure a fair and rigorous comparison, all baseline methods were trained using the data augmentation protocol (e.g., random crops, flips) as our proposed method. However, different methods may prefer different augmentation strengths, and we report the best performance achieved for each. We will add a clarifying sentence to the experimental setup section to make this explicit and avoid ambiguity.

---

> > ### Comment · Reviewer_JCUz · 2025-08-07
> >
> > Thank you for the rebuttal. After reading the authors' response, I intend to maintain my rating.

---

> ### Author Response · Authors · 2025-08-09
> **Thank you**
>
> Thank you again for your thoughtful comments. If you have any further comments, we will gladly make every effort to address them.

---

### Official Review · Reviewer_Z7dt · 2025-07-06

**Clarity:** 1
**Significance:** 3
**Originality:** 3
**Rating:** 4
**Confidence:** 5

**Summary:**

The authors introduce a novel DRO framework where (semi-relaxed and entropy-regularized) Optimal Transport is utilized together with (semi-relaxed) Inverse Optimal Transport during the DRO optimization to find a robust model/classifier. It is argued that the ability of such models to withstand various realistic distribution shifts, while showing high source accuracy, depends critically on the support \nu of the computed (uncertainty) distributions Q. The authors then, inspired by the proposed Inverse Optimal Transport which refines the computed couplings of the semi-relaxed OT step, propose an adaptive filtering operation to refine this reference distribution \nu and hence shape the geometry of the computed Q_{\lambda} distributions. Results are shown, over various DRO benchmarks, where it is shown that the proposed method can improve over state-of-the-art approaches, such as Sinkhorn DRO.

**Questions:**

- It would be good to expand on the Figure 1 and illustrate why and how the proposed method can find better / more meaningful / realistic uncertainty sets (through semantic calibration / adaptive filtering etc.)
- It would be clearer in the discussion that follows eq. (1) to mention that \nu forms the support of the distributions Q.
- "This choice is both practical and theoretically sound, as the relative entropy term then differs only by a constant when Ptr ≪ μ leaving the optimization unaffected."
This sentence was not clear to me.
- The paper becomes sometimes quite confusing to read and to understand in section 3, partly due to the fact that that the authors do not separate the problem formulation from the contributions/to be introduced method. For instance the 'Distribution Exploration via OT' paragraph can be streamlined with section 2 where the authors introduce the derivation of eq. (6) from Wang et al. [58].  The following Lemma 3.1 should also be mentioned to follow from [58].  Also, the 'classification as inverse OT' part in section 3.1 is not clear as suddenly the OT problem changes from matching p(x) and q(x) to p(x) and y.
- line 157: lambda depends on delta, which is not mentioned.
- It would make section 3 clearer if Remark 1 is explained more explicitly: show formally how (1) can be cast as OT + IOT mentioned in section 3.1
- It is not clear what is the benefit of emphasizing 'semantic calibration' with 'inverse OT' over plain source risk minimization using cross-entropy. As one trains the model f_theta on the distributions Q, the parameters theta changes the OT costs and there doesn't seem to be any additional benefit of formulating this problem as an 'inverse OT'. For instance the 'calibration' of costs via IOT in (11) and (12) can I think be equally achieved by ERM that includes augmentations to learn a better model g_augmented, which would then improve the semantic quality of costs C(g_augmented(x_i), g_augmented(x_j)) over C(g(x_i), g(x_j)).
- "Leveraging this insight, we define the label-level transport cost as..." The preceding discussion for motivating (14) is not clear for me.
- Given the heuristic nature of the introduced filtering operations for \nu in (15), (16) I have a hard time understanding the reasoning behind them. Since this part seems to form the core of the paper's contributions, e.g. over [60]: "Outlier-robust distributionally robust optimization via unbalanced optimal transport", it would make sense I think to explain, motivate and analyze them, as well as the connections to other thresholding strategies such as FreeMatch, in much more detail. When do these heuristics work well and when would they not for instance?
- In the discussion that follows Thm 3.2, it would be nice to mention similar results for other DRO algorithms (e.g., ones compared to in the experiments). Is Thm 3.2 an extension of such results for dynamically evolving uncertainty sets?
- It is not clear why Algorithm 1 needs step 1 [train \theta by the joint objective in (12) and (13) for N0 epochs] before the for loop.

Experiments
- Since they include augmentations, the authors should compare against ERM with augmentations included. The other methods compared against should then also include augmentations for fairness.
- I suspect that the proposed method may not beat the 'ERM with augmentation' baseline across various distribution shift scenarios. See the "In search of lost domain generalization" paper and the DomainBed platform for why I think so. Would it be possible to try more algorithms or datasets from the DomainBed repository?
- GIW is not evaluated in the experiments, nor should it be: in DRO one does not assume unlabeled test/target distribution to be available and domain adaptation methods such as GIW require such data to be available. However it is mentioned as if it is compared against.
- The authors should describe the distribution shift in each scenario: e.g., in colored MNIST, the 'worst group' do not contain the colors correlated with the labels, I believe. What is the difficulty in the Waterbirds dataset? Do the spurious correlations (background to label correlation) change in the 'worst-group' distribution?
- In general the experiments section, even when taken together with Appendix D, is quite inadequate to appreciate the contributions. The authors should streamline parts of Section 3 with Section 2 to create more space in the main text, so that they can significantly widen the
discussion of the experiments section.
- Hyperparameters set during the experiments should be mentioned in particular, e.g., what is the value of the uncertainty set radius \delta
  in the experiments? Is it varying or constant?
- "Notably, when the correlation between color and label is strong (bias = 0.9), all other DRO methods suffer a significant drop except AdaDRO, which effectively mitigate the over-pessimism nature of DRO" Which experiment is this? Show the quantitative effect.
- Report error bars, at least in the appendix.
- "We conduct ablation experiments to isolate the effect of AdaDRO’s technical components, please refer to the supplementary materials"
Please include all relevant material in the Appendix.
- "Full experimental details and additional results, including the visualization of the dynamic \nu_t sensitivity analyses, and runtime statistics, are available in the supplementary materials."
Please include some visualizations and other analyses/statistics in the appendix / experiments section also. Ideally, there shouldn't be any need to include a supplementary material besides the appendix.
- How do the different imbalance factors effect the CIFAR-10/100-LT experiments?
- Which experiment/dataset does Table 6 correspond to?
- In Table 8, "Higher λ corresponds to smaller uncertainty sets. AdaDRO maintains stable performance, while Sinkhorn DRO suffers significantly as λ increases"
Higher \lambda should correspond to bigger uncertainty sets (bigger radius \delta) I believe. The conclusion also seems unwarranted: according to the results, there is similar variance between the two methods.

Minor comments:
- W matrix is R^{d_e \cross K} instead
- note that eq. (2) is the KL-divergence between \gamma and \mu \cross
  \nu whereas eq. (3) is the \emph{negative} entropy H plus some
  constant. Hence it would be more appropriate to use KL-notation for
  eq. (2) and show that it is equal to the negative of Shannon entropy
  H(.) up to a constant.  Minimizing the KL-divergence in this case
  maximizes the Shannon entropy and vice versa.  Regularizing with the
  KL-divergence of a coupling likewise inflates the entropy of the
  coupling, depending on the regularization constant.
- not a big fan of the p,q notation for (x,y) positions as they're easily
  confused with the upper case notation for the distributions
- line 151: add space after 'following'
- 'Geometric Awareness' is not a benefit of the entropy regularization
  in D_{\epsilon} but rather a 'benefit' of the OT-formulation.
- The inputs xi and yi generally live in different vector spaces (with
  diff. dimensions), so the coupling gamma_bar in (8) is not standard
  for Wasserstein distances (Gromov-Wasserstein OT considers points
  that can live in different spaces.) You'd need to introduce the
  operation of the model f_theta: x -> y_hat with parameters theta
  inside Q^{\lambda} to make it compatible with standard OT
  formulation.
- N0 is also a required hyperparameter of Alg1

**Ethical Concerns:**

["NO or VERY MINOR ethics concerns only"]

**Final Justification:**

I thank the authors for their carefully prepared rebuttal. After going through the response, I have no major concerns and advocate for the acceptance for this paper. I have raised my overall score (and confidence) accordingly.

A few issues still remain, although they are not as critical:

- ERM with augmentation is still competitive or beating the proposed algorithm in the DomainBed benchmark that is posted by the authors, except for the Colored MNIST experiments. Given the response (to Q1-Q2) where the authors claim that their method would be better due to (i) label noise, (ii) label-agnostic calibration, I infer that (i) label-noise is probably not prevalent in a lot of these 'curated' datasets (is that so?), and that (ii) label-agnostic calibration indeed helps to avoid spurious correlations in a way that ERM with augmentation cannot. I would like the authors to discuss these experiments and these (or similar) conclusions in more detail.
- The heuristic for the nu-filtering should be explained better and integrated with Figure 1 (which the authors kindly agreed to do). It would be quite nice to also discuss potential problems with this heuristic.
- It is not clear from the main text what is the precise novelty of Theorem 3.2 and what is the related work. The contributions of the paper towards this result should be written and highlighted clearly in the main text.

**Limitations:**

"If the authors answer NA or No, they should explain why their work has no societal impact or why the paper does not address societal impact."
It would be good to briefly mention the positive societal impacts.

**Quality:**

3

**Strengths And Weaknesses:**

The authors identified the underlying dilemma of the DRO problem quite correctly and proposed an interesting solution to solve this important issue, which prevents the wider deployment of DRO methods in practice (as the proposed solutions are either too conservative and have poor source accuracy or are unable to capture the relevant uncertainty set). However, the proposed solution is a heuristic that is not explained or motivated carefully. I also found some parts of section 3 to be unclear in some aspects (as explained below in detail), which prevented me from appreciating the proposed method and the resulting Algorithm 1. Equally importantly, the experiments sections was quite limited and the results were not explained carefully. A much more detailed experiments section with more results (over more distribution shift scenarios) and ablations, would make a more convincing case. Please find below some detailed comments concerning both these issues.

As of now I cannot advocate for acceptance of this paper in its current form, but if the authors can answer my concerns adequately I'd be happy to raise my score.

---

> ### Author Rebuttal · Authors · 2025-07-31
>
> We thank the reviewer for the thoughtful comments, and address your concerns below.
>
> **Q1 - Q6:**
>
> We sincerely thank the reviewer for their detailed and constructive feedback on the manuscript's presentation. We agree that the clarity and structure can be improved.  In particular, we will enhance Figure 1 by including additional visualizations to better illustrate how our semantic calibration and adaptive filtering contribute to the formation of more meaningful uncertainty sets. We will also revise the text to explain Remark 1 more explicitly. Furthermore, we will incorporate additional details as suggested.
>
> **Q3:** "This choice is both practical and theoretically sound, as ...
>
> $H(\gamma | \mu \otimes \nu) = E_{(p,q)\sim\gamma} [ \log(\frac{d\gamma(p,q)}{dP_{tr}(p)d\nu(q)}) + \log(\frac{dP_{tr}(p)}{d\mu(p)}) ]  = E_{\gamma} [ \log(\frac{d\gamma(p,q)}{dP_{tr}(p)d\nu(q)}) ] + E_{p\sim P_{tr}}[\log(\frac{dP_{tr}(p)}{d\mu(p)}) ]$. Thus
> any choice of $\mu$ satisfies $P_{tr} \ll \mu$ yields an equivalent entropy term up to an additive constant, and does not affect the optimization.
>
>
> **Q4:** ... the 'classification as inverse OT' part... the OT problem changes from matching $p(x)$ and $q(x)$ to $p(x)$ and $y$.
>
> Our contribution in Lemma 3.1 is to provide a key reinterpretation for DRO formulated in (1): we show that the worst-case distribution (derived in [1]) can be understood as the target marginal of a specific semi-relaxed OT problem. This perspective forms the foundation for our subsequent components—semantic calibration via inverse OT (IOT) and adaptive filtering—both of which exploit the geometric structure of OT.
> Regarding the discussion of ‘classification as inverse OT’, we adapt the IOT framework between the source distribution of training data $p(x)$, and the target distribution over labels $p(y)$. This is a necessary step for modeling classification as a transport-based matching.
>
>
> **Q7:** ... the benefit of emphasizing 'semantic calibration' with 'inverse OT' ...
>
> As described in lines 102 and 209, the transport cost between two samples is decomposed into two parts: a sample-level cost and a label-level cost,
> $C(p_i, p_j) = C^X(x_i, x_j) + C^Y(y_i, y_j)$.
> Our semantic calibration framework explicitly addresses both components using the IOT objectives (12) and (13), respectively. We reinterpret the cross-entropy loss through the lens of IOT and show that it coincides with our objective in (13). Importantly, this perspective does not complicate the formulation, and provides a deeper understanding of its function within our IOT framework.
>
> In contrast, standard ERM does not provide well-founded mechanism for learning the sample-level cost $C^X(x_i, x_j)$, and thus cannot guarantee that the learned feature space reflects meaningful semantic geometry—even with data augmentation.
> To address this, we introduce the IOT objective (12), which explicitly learns $C^X$ by recovering latent semantic distances from observed matching pairs. This approach helps to extract $C^X$, enabling the semantic alignment that cannot be achieved by ERM alone.
>
>
> **Q8:** ... preceding discussion for motivating (14) is not clear for me.
>
> We will revise this section to improve clarity. The high-level idea is as follows: standard cross-entropy training is equivalent to an instance of IOT, where the training objective effectively calibrates the distance defined in (8) between the input embedding $g_\theta(x_i)$ and its corresponding class weight vector $w_{y_i}$.
> This process naturally yields a set of weight vectors $\{w_k\}_{k=1}^K$ that carry semantic meaning; classes that are semantically similar will have similar weight vectors. Within the IOT framework, these weights induce a transport cost over the label space $C^Y$.
> We will present this idea more directly in the revision.
>
>
> **Q9:** ... I have a hard time understanding the reasoning...
>
> The key idea of the filtering is to use the model's own confidence (i.e., the coupling $\gamma_\theta$) as a self-adaptive signal. If the model has difficult to align a sample $p$ with its ground-truth match $q$ (i.e., $\gamma_\theta(p,q)$ is low), it suggests that $p$ may be an outlier or noisy sample that is harming the uncertainty set.
> **Adaptive Threshold $\tau(q)$:** The adaptive threshold is designed to balance global and local confidence. The term $\tau_1$ is the global average confidence, setting a baseline. $\tau_2(q)$ measures the "matchability" of a target $q$. The ratio "$\tau_2(q) / \max(\tau_2)$" adjusts the global threshold locally. For easy targets (high $\tau_2$), the threshold is higher; for hard targets (low $\tau_2$), the threshold is lower to avoid incorrectly filtering. The popular FreeMatch [2] used a similar self-adaptive thresholding, and was later also reinterpreted through the lens of OT [3].
>
>
> **Q10:** ... mention similar results... Is Thm 3.2 an extension of such results ...?
>
> Thank you for prompting us to highlight this point.
> In addition to handling the dynamically evolving reference distribution $\hat{\nu}$, our analysis addresses a more general non-convex setting. As far as we know, the most relevant prior result is provided in [1]. While [1] focuses on convex settings, our analysis extends to non-convex regimes with dynamically evolving uncertainty sets, using different techniques.
>
>
> **Q11:**  ... why Algorithm 1 needs step 1 ...
>
> Step 1 acts as a warm-up phase to ensure stable and reliable signals for use in (12) and (13).
>
>
> **Experiments**
>
> **Q1 Q2: ... 'ERM with augmentation' baselines ...**
>
> The refereed paper show that ERM+augmentation is a strong baseline.
> We believe our method is unique in two aspects: 1, its label-agnostic sample-level semantic calibration (13); 2, the filtering mechanism based on the coupling introduced by (12) (13).
>
> An intuitive example is the presence of noisy labels: when a portion of labels are flipped, sample-level calibration (12) can still reliably capture semantics. This is because the process is unsupervised, similar to contrastive learning [4], and thus remains robust to label noise. Building on this, the coupling-based filtering mechanism can discards samples with wrong label, enabling the cross-entropy to train on a cleaner subset of the data.
> In contrast, methods that depend on label supervision, including ERM with augmentation, face an inherent "garbage in, garbage out" issue. No  augmentation can correct mislabeled examples, and these label errors inevitably propagate through the learning process, compromising model robustness.
>
> A similar advantage is observed in Colored MNIST. Even when augmentation occasionally alters the spurious attribute (e.g., color), standard ERM still learns entangled representations, as the label remains strongly correlated with color. In contrast, our label-agnostic calibration encourages the model to identify truly invariant features—those that remain stable under augmentation—naturally prioritizing shape over color in the absence of a strong supervisory signal.
>
> For the space limit, we only present results on the DomainBed benchmark.
>
> | Algorithm | CMNIST         | RMNIST         | VLCS           | PACS           | Office-Home    | TerraInc       | DomainNet      |
> | --------- | -------------- | -------------- | -------------- | -------------- | -------------- | -------------- | -------------- |
> | ERM+Aug   | 52.0 ± 0.1     | **98.0 ± 0.0** | 77.4 ± 0.3     | 85.7 ± 0.5     | **67.5 ± 0.5** | 47.2 ± 0.4     | **41.2 ± 0.2** |
> | Ours      | **63.8 ± 0.3** | **98.0 ± 0.1** | **77.8 ± 0.3** | **86.2 ± 0.6** | 67.1 ± 0.3     | **50.2 ± 0.5** | 39.7 ± 0.3     |
>
> **Q3-Q13:** We thank the reviewer for sufficient suggestions.
> We will clarify these points in the revised experimental section:
> (Q3) GIW was mentioned for context and we will make it clear that it was not used as a baseline; (Q4) For Waterbirds, the spurious correlation is between the background (water vs. land) and the label (waterbird vs. landbird). The 'worst-group' distribution contains samples where this correlation is broken (e.g., waterbirds on land); (Q6) The uncertainty radius $\delta$ is a hyperparameter, which is controlled by the dual variable $\lambda$ in practice. We will add its search space and the selected value for each experiment to the appendix; (Q12) Table 6 corresponds to the Waterbirds dataset. We will add a caption.
>
> We agree with these excellent suggestions for improving the paper's readability and completeness. In the revision, we will:
> (Q5) Streamline Sections 2 and 3 to create more space for a detailed discussion of the experiments; (Q8, Q9, Q10) Integrate all supplementary materials, including ablation studies, visualizations, error bars, and full experimental details, directly into the main paper's appendix to create a single, self-contained document; (Q11) Add a discussion on the effect of imbalance factors in the CIFAR-LT experiments.
>
> **Response to Minor Comments:** Thank you for your valuable comments. We have corrected the typos and made several revisions to improve the overall writing quality of the manuscript.
>
> Regarding the comment on the inputs $\textit{x}_i$ and labels $\textit{k}$ residing in different spaces, our formulation does not compute a  Wasserstein distance, which is defined in a metric space. Instead, our approach leverages the more general OT framework, which permits flexible ground distance function that  do not necessarily satisfy the properties of a metric. As long as a ground distance function is defined, an OT distance can be computed.
>
> [1] Sinkhorn distributionally robust optimization. 2023
>
> [2] Freematch: Self-adaptive thresholding for semi-supervised learning. ICLR'23
>
> [3] OTMatch: Improving Semi-Supervised Learning with Optimal Transport. ICML'24
>
> [4] Understanding and Generalizing Contrastive Learning from the Inverse Optimal Transport Perspective. ICML'23

---

> > ### Comment · Reviewer_Z7dt · 2025-08-07
> > **thanks**
> >
> > Thank you for your carefully written rebuttal and sorry for the late response. I have changed my score, please look at the 'Final Justification' area for some final comments also.

---

> > > ### Author Response · Authors · 2025-08-09
> > > **Thank you**
> > >
> > > Thank you again for your thoughtful comments. If you have any further comments, we will gladly make every effort to address them.

---

> ### Comment · Area_Chair_JcDA · 2025-08-06
>
> Dear reviewer
>
> Please engage in the discussion with the authors at your earliest convenience.
>
> Thanks
>
> AC

---

### Comment · Area_Chair_JcDA · 2025-08-02

Dear Reviewers,

Please take a look at the authors' response and discuss if there are still more concerns that need to be clarified.
Thanks
AC

---

### Note · Authors · 2025-08-15

We sincerely thank all reviewers for their insightful feedback. We are pleased that the reviewers recognized our work's key strengths:

- **Insightful framework and innovative techniques:** Our work reframes entropy-regularized DRO as a dynamic exploration-and-alignment process, offering both *geometric intuition* and *theoretical clarity* for a challenging problem. This perspective naturally gives rise to novel techniques supported by rigorous theoretical guarantees.
- **Promising experimental results**: Experiments across diverse datasets consistently demonstrate improvements over strong baselines.

During the discussion, we aimed to address all major concerns:

- **Computational efficiency:** We clarified that our method employs a *closed-form solution* by adopting a semi-relaxed OT formulation that enforces only the source marginal constraint. This design avoids costly iterative solvers, ensuring scalability with minimal overhead. This addressed the efficiency concerns raised by Reviewers sEAy and YPiL.
- **Expanded experiments & ablations:** In response to reviewers requests, we added a portion of new results on the DomainBed benchmark and ImageNet-LT, along with additional ablation and sensitivity analyses. We will incorporate the full set of new results and visualizations in the revised version.  Notably, both empirically and intuitively, our label-agnostic semantic calibration (Eq. 13) and coupling-based filtering offer clear advantages in the presence of noisy labels or strong spurious attributes—settings where ERM, even with advanced data augmentation, often fails to mitigate adverse effects.
- **Comprehensive Revisions for Clarity and Completeness**:  We thank the reviewers for their insightful suggestions. We revised the manuscript for clarity by streamlining the presentation to better motivate our contributions and expanding the experimental section with more detailed analyses. To provide a more complete picture of our work, the revision also includes a dedicated discussion on limitations and integrates all supplementary materials into a self-contained appendix.

We sincerely thank the reviewers again for their time, constructive comments, and efforts in improving our work.

---

### Decision · Program_Chairs · 2025-09-17

**Decision:**

Accept (poster)

**Comment:**

This paper develops a new DRO framework where Optimal Transport is utilized together with (semi-relaxed) Inverse Optimal Transport during the DRO optimization to find a robust model/classifier. The proposed approach addresses the current limitation of DRO approach, which is the design of the uncertainty set. Instead, the authors develop a novel perspective that casts entropy-regularized Wasserstein DRO as a dynamic process of distributional exploration and semantic alignment. Theoretical convergence guarantees were also provided by the authors. All the reviewers are positive about this manuscript and appreciate its novelty and significance for improved robustness over existing benchmarks. Some minor concerns, although not critical, still exist. The authors may address them in a future version of this paper.